# Thermodynamically Consistent Models for Coupled Bulk and Surface Dynamics

**DOI:** 10.3390/e24111683

**Published:** 2022-11-17

**Authors:** Xiaobo Jing, Qi Wang

**Affiliations:** 1Beijing Computational Science Research Center, Beijing 100193, China; 2Department of Mathematics, University of South Carolina, Columbia, SC 29028, USA

**Keywords:** thermodynamically consistent model, binary phase field model, dynamic boundary conditions, energy dissipation

## Abstract

We present a constructive paradigm to derive thermodynamically consistent models coupling the bulk and surface dynamics hierarchically following the generalized Onsager principle. In the model, the bulk and surface thermodynamical variables are allowed to be different and the free energy of the model comprises the bulk, surface, and coupling energy, which can be weakly or strongly non-local. We illustrate the paradigm using a phase field model for binary materials and show that the model includes the existing thermodynamically consistent ones for the binary material system in the literature as special cases. In addition, we present a set of such phase field models for a few selected mobility operators and free energies to show how boundary dynamics impart changes to bulk dynamics and vice verse. As an example, we show numerically how reactive transport on the boundary impacts the dynamics in the bulk using a reactive transport model for binary reactive fluids by adopting a structure-preserving algorithm to solve the model equations in a rectangular domain.

## 1. Introduction

Thermodynamically consistent models refer to the ones derived from thermodynamical laws and principles. In particular, they obey the second law of thermodynamics. The generalized Onsager principle (GOP) is a protocol in which the Onsager linear response theory combined with the equilibrium maximum entropy principle is extended to describe thermodynamically consistent models comprising both reversible and irreversible processes [1,2,3,4]. The generalized Onsager principle warrants the second law of thermodynamics in the form of the Clausius–Duhem inequality and has proven to be an effective theoretical framework for developing thermodynamically consistent models at various time and length scales [3,5,6,7,8]. In the past, the generalized Onsager principle and the equivalent thermodynamical second law has been primarily used to derive dynamical equations in the bulk while boundary conditions are largely trivialized by assuming adiabatic, static boundary conditions, or periodic boundary conditions. In many problems where the scale in the bulk matches the scale in the boundary in terms of dynamics, the dynamics at the boundary can no longer be overlooked or trivialized. For material systems confined in a compact domain with active boundary dynamics, materials properties in the bulk and those on the boundary surface may differ significantly. As a result, different order parameters may need to be introduced to best describe their respective dynamics. This is an issue that has not been paid much attention in the literature. In this study, we revisit these issues and present a general, hierarchical framework to derive thermodynamically consistent models for material systems with coupled dynamics in the bulk and at the boundary using the constructive, generalized Onsager principle [1]. We illustrate the gist of the approach by deriving thermodynamically consistent phase field models and consistent boundary conditions for binary materials in a smooth domain owing to the popularity of phase field models in the literature. However, the theoretical framework is by no means limited to the phase field models.

Phase field modeling is one of the powerful and versatile modeling paradigms for studying multi-phasic materials in domains with complex interfacial geometries and complex interfacial phenomena between distinct phases (immiscible mixtures) or even miscible mixtures [9,10,11]. It is especially useful and effective when handling dynamical phase boundaries in multi-phasic materials involving topological changes compared to other methods, such as front tracking methods, level-set methods, volume-of-fluid methods, etc. [11,12,13]. By design, it is for diffuse interfaces with certain interface thicknesses in which complex interfacial dynamics prevail. A quality phase field model should be able to capture well-known sharp interface conditions (e.g., Gibbs–Thomson condition) in the case of vanishing thickness of the interface in immiscible mixtures [14,15]. This requires one to be mindful when deriving the free energy of the system in the phase field model so that thermodynamical laws and principles are followed faithfully. Notice that the advantage of the phase field model in dealing with diffuse interfaces may also limit its applicability to resolve sharp interfaces in light fluids [16]. The Allen–Cahn and Cahn–Hilliard equations are two classical phase field models used to describe dynamics in the multi-phasic systems and both of them share the same chemical potential if the free energy functionals are the same. However, the Allen–Cahn equation does not preserve the conservation of the bulk volume while the Cahn–Hilliard equation does [17,18,19]. In life science, materials science, and many engineering fields, there is an abundance of multi-phasic material systems with diffuse interfaces, which have kept multi-phase field models popular and practical [12,20,21,22].

There are quite a number of phase field models in the literature today. However, not all are thermodynamically consistent. Even for the thermodynamically consistent ones, the proper boundary conditions are not well studied until recently. In particular, when one considers surface dynamics, an issue emerges right away that is what is the relation between the phase field variable in the bulk and the one on the boundary? By assuming these two are the same at the boundary, a series of studies have examined the issue of thermodynamically consistency [23,24,25,26,27]. By not assuming that they coincide on the boundary, Knopf et al. derived a set of boundary conditions for the Cahn–Hilliard model (known as the Knopf–Lam model) in 2020 [28] and for non-local models (known as the Knopf–Signori model) in 2021 [29]. The latter studies opened up a new venue for one to examine the thermodynamically consistency in the bulk and surface dynamics holistically. To make the paper more readable, we list some of the existing models in the literature first. In Section 3, we will show that all these existing models are the limiting cases of the general model we proposed.

Jing–Wang model [24]:We firstly define the free energy of the system as follows
(1)$$\begin{array}{c} \begin{array}{c} E\left[\phi \right]=\int_{\Omega }e_{b}\left(\phi ,\phi_{t},\nabla \phi ,\nabla \nabla \phi \right)dx+\int_{\partial \Omega }\left[e_{s}\left(\phi ,\phi_{t},\nabla_{s}\phi ,\nabla_{s}\nabla_{s}\phi \right)\right]ds, \end{array} \end{array}$$
where ϕ is the order parameter, $\nabla_{s}$ is the surface gradient. The corresponding governing equation system is given by
(2)$$\begin{array}{c} \begin{array}{c} \phi_{t}=-\left(M_{b}^{\left(1\right)}-\nabla \cdot M_{b}^{\left(2\right)}\cdot \nabla \right)\mu_{b},\;x\in \Omega ,\\ \phi_{t}=-\left(M_{s}+\frac{\beta^{2}}{\alpha }\right)\left(\mu_{s}+\mu_{c}\right)+\frac{\beta }{\alpha }\mu_{b},\;s\in \partial \Omega ,\\ \alpha n\cdot M_{b}^{\left(2\right)}\cdot \nabla \mu_{b}=-\mu_{b}+\beta \left(\mu_{s}+\mu_{c}\right),\;\nabla_{n}\phi_{t}=-M_{g}\mu_{g},\;s\in \partial \Omega , \end{array} \end{array}$$
where $M_{b}^{\left(1\right)}$, $M_{b}^{\left(2\right)}$ and $M_{s}$ are mobilities, α and β are parameters, $\mu_{b}$ and $\mu_{s}$ are chemical potentials, and $\mu_{g}$ is the conjugate variable, which are defined in Section 2.Knopf–Lam model [28]:We define the free energy as follows
(3)$$\begin{array}{c} \begin{array}{c} E=\int_{\Omega }\frac{\epsilon }{2}{\left|\nabla \phi \right|}^{2}+\frac{1}{\epsilon }F\left(\phi \right)dx+\int_{\partial \Omega }\frac{\sigma }{2}{\left|\nabla_{s}\psi \right|}^{2}+\frac{1}{\sigma }G\left(\psi \right)+\frac{1}{2K}{\left(H\left(\psi \right)-\phi \right)}^{2}ds, \end{array} \end{array}$$
where ϕ and ψ are two order parameters to describe the materials in the bulk and on the boundary surface, respectively. ϵ, σ, and *K* are model parameters, H(ψ) is a function of ψ. The governing equations in the bulk and on the boundary are given by
(4)$$\begin{array}{c} \begin{array}{c} \phi_{t}=\nabla^{2}\mu_{b},\;\mu_{b}=-\epsilon \nabla^{2}\phi +\frac{1}{\epsilon }F^{\prime }\left(\phi \right),\;x\in \Omega ,\\ \psi_{t}=\nabla_{s}^{2}\mu_{s},\;\mu_{s}=-\sigma \nabla_{s}^{2}\psi +\frac{1}{\sigma }G^{\prime }\left(\psi \right)+\epsilon H^{\prime }\left(\psi \right)n\cdot \nabla \phi ,\;s\in \partial \Omega ,\\ n\cdot \nabla \mu_{b}=0,\;\epsilon n\cdot \nabla \phi =\frac{1}{K}\left(H\left(\psi \right)-\phi \right),\;s\in \partial \Omega . \end{array} \end{array}$$Liu–Wu model [27]:By setting H(ψ)=ψ and K→0, H(ψ)→ϕ in the Knopf–Lam model, the Liu–Wu model is obtained which are given by
(5)$$\begin{array}{c} \begin{array}{c} \phi_{t}=\nabla^{2}\mu_{b},\;\mu_{b}=-\epsilon \nabla^{2}\phi +\frac{1}{\epsilon }F^{\prime }\left(\phi \right),\;x\in \Omega ,\\ \phi_{t}=\nabla_{s}^{2}\left(\mu_{c}+\mu_{s}\right),\;\mu_{c}+\mu_{s}=\epsilon n\cdot \nabla \phi -\sigma \nabla_{s}^{2}\phi +\frac{1}{\sigma }G^{\prime }\left(\phi \right),\;n\cdot \nabla \mu_{b}=0,\;s\in \partial \Omega . \end{array} \end{array}$$Knopf–Signori model [29]:Once the bulk and surface free energies are non-local as in Section 3.3 below, the non-local dynamics are given by
(6)$$\begin{array}{c} \begin{array}{c} \phi_{t}=M_{b}^{\left(2\right)}\nabla^{2}\mu_{b},\;x\in \Omega ,\\ \psi_{t}=-\left(-M_{s}^{\left(2\right)}\nabla^{2}+\frac{\beta^{2}}{\alpha }\right)\mu_{s}+\frac{\beta }{\alpha }\mu_{b},\;s\in \partial \Omega ,\\ \alpha M_{b}^{\left(2\right)}n\cdot \nabla \mu_{b}=-\mu_{b}+\beta \mu_{s},\;s\in \partial \Omega . \end{array} \end{array}$$

In this study, we focus on the demonstration of the paradigm through the derivation of thermodynamically consistent phase field models with coupled surface and bulk dynamics of potentially distinct phase field variables, which include the derivation of the transport equation in the fixed bulk domain, as well as the consistent one on its boundary. We will present the models with free surface domains and boundaries in a sequel. We stress that it is important to study multi-phase material systems using a thermodynamically “correct” model since it not only gives one a comprehensive description of the “correct” physics for the material system but also gives one a well-posed mathematical system to analyze and compute. Speaking of a thermodynamically “correct” model, we insist that the model must be at least thermodynamically consistent and obeys known physical laws. This humble criterion would perhaps disqualify a host of existing phase field models. In addition, we notice that most of the studies on phase field models are concentrated on equations in the bulk with static or periodic boundary conditions at fixed boundaries, where the boundary contributions to thermodynamical consistency are trivialized.

Given the recent technological advances in materials science and engineering and the abundance of natural phenomena where boundaries regulate the internal dynamics, boundaries of a material-confining device can no longer be treated as passive. They can be made with distinctive properties to interact or even control the material within the device, especially, in thin films [30,31,32]. For instance, the newly discovered boundary effect to the existence of blue phases in cholesteric liquid crystals in microscales across a quite large temperature range is one of the prominent examples [33,34,35]. In life science, dynamics in cell membranes play a significant role in determining cell behavior. These require one to derive a model for the material system confined in a compact domain to take into account the potential dynamical contribution from the boundary. There has been a surge in activities in this direction on phase field models and reaction-diffusion models recently [36,37,38,39,40,41,42].

In this paper, the idea is exemplified through a derivation of a phase field model. There in fact exists an underlying paradigm for deriving much wider classes of thermodynamically consistent models in fixed domains with smooth boundaries, whose thermodynamical variables in the bulk may differ from the ones at the boundary and the free energy depends on gradients of the variables up to the second order or strongly non-local. In this approach, we begin with the prescribed total free energy comprising the bulk free energy, surface free energy and free energy that couples the bulk and surface order parameters or phase field variables and calculate the energy dissipation rate of the total free energy. The fact that order parameters in the bulk and ones at the boundary can be different in the continuum level description of the thermodynamical system is common. For example, the order parameter in the bulk in the phase field model is the volume or mass fraction, while the one at the boundary is the area fraction. Their definitions are different so that they may not be the same while evaluated at the boundary. Thus, it is quite common that the surface dynamics and the dynamics in the bulk are described by different thermodynamical variables, dictated by the respective materials properties of the bulk and the surface. In this case, in addition to the bulk and surface energy, an additional energy, defined as the coupling energy, must be introduce to formulate the continuum theory.

In the derivation, we first apply the constructive, generalized Onsager principle to the bulk energy dissipation rate to arrive at constitutive equation system that warrants the bulk energy dissipation. We note that the mobility operator stipulated in the bulk constitutive equation may affect the boundary energy transport. Secondly, after accounting for the additional surface energy dissipation due to the non-local mobility operator in the bulk, we apply the generalized Onsager principle subsequently to the surface energy contribution to the total energy dissipation. This yields the constitutive equation for surface transport at the boundary. Combining the results from both steps, we arrive at coupled bulk and surface transport equation systems that yield a coupled, thermodynamically consistent model. We remark that this paradigm applies to any thermodynamical system. To derive hydrodynamical systems, however, additional conservation laws must be enforced in addition to the constitutive equations, which we will defer to another paper.

The paper is organized as follows. In Section 2, we present the paradigm using a phase field model for a binary system as an example and discuss its various limits. In Section 3, we discuss some special cases, the strongly non-local free energy with dynamic boundary conditions, and a reactive transport system. In Section 4, we demonstrate the effect of dynamic boundary conditions on the solution in the bulk using energy-dissipation rate preserving numerical simulations. We summarize the results in Section 5.

## 2. Thermodynamically Consistent Phase Field Models with Consistent Dynamic Boundary Conditions

We illustrate the general framework for deriving transport equations in the bulk and consistent dynamic equations on the boundary for a thermodynamical model that yields a negative energy dissipation or a positive entropy production rate, using a scalar phase field model for a binary material system with free energy consisting of up to second order spatial derivatives of the phase field variable. Then, we elucidate the path for extending it to the more general free energy functional including the non-local free energy for general thermodynamical models. We will show how existing phase field models of this kind are deducible from the general framework. We focus on the derivation in the isothermal case here so that the free energy is the proper thermodynamical potential to work with.

### 2.1. Generalized Onsager Principle

The classical Onsager linear response theory on which the Onsager principle for dissipative systems is based provides a viable way to calculate dissipative forces in relaxation dynamics in an irreversible non-equilibrium process [2,4,43,44]. In the general setting, the linear response theory states that given a chemical potential in an isothermal system, the generalized flux $\phi_{t}$ is proportional to the generalized force or chemical potential μ [17,18,19] as follows
(7)$$\begin{array}{c} \phi_{t}=-M\mu , \end{array}$$
where *M* is the mobility operator. For dissipative systems where dynamics are irreversible, the additional Onsager reciprocal relation dictates that *M* is symmetric; for conservative systems where dynamics are reversible, *M* is antisymmetric [1]. In general, *M* comprises the symmetric part $M_{s}$ and antisymmetric part $M_{a}$: $M=M_{s}+M_{a}$. We note that when *M* is a differential operator, such as in the Cahn–Hilliard equation system, the symmetric property of *M* is also determined by the boundary conditions of the system as well. For a system where inertia is non-negligible and there coexist irreversible and reversible dynamics in the non-equilibrium process, we extend the force balance equation to a generalized Onsager principle [1,24]
(8)$$\begin{array}{c} -M^{-1}\phi_{t}=\rho \phi_{tt}+\mu =\tilde{\mu }\Leftrightarrow \phi_{t}=-M\left(\rho \phi_{tt}+\mu \right)=-M\tilde{\mu }, \end{array}$$
where $\tilde{\mu }$ is the general chemical potential, $\rho \phi_{tt}$ represents the inertia force, and ρ is a measure of mass. We next use the generalized Onsager principle to derive the general phase field model along with its consistent boundary conditions for a binary material system. For convenience, we omit $\tilde{\left(•\right)}$ over variable (•) in the following.

### 2.2. Models with the Free Energy up to Second Spatial Derivatives

Let the bulk free energy in a fixed material domain Ω be given by
(9)$$\begin{array}{c} E_{b}\left[\phi \right]=\int_{\Omega }e_{b}\left(\phi ,\phi_{t},\nabla \phi ,\nabla \nabla \phi \right)dx, \end{array}$$
where $e_{b}$ is the energy density per unit volume. Especially, the kinetic energy in the bulk to account for the inertia effect in the system is included in $e_{b}$, which is related to $\phi_{t}$ and is usually be chosen as $\frac{\rho }{2}\phi_{t}^{2}$. We consider a binary material system with a boundary that may have its distinctive properties than the bulk and possesses its own surface energy of derivatives up to the second order in space
(10)$$\begin{array}{c} E_{s}\left[\psi \right]=\int_{\partial \Omega }e_{s}\left(\psi ,\psi_{t},\nabla_{s}\psi ,\nabla_{s}\nabla_{s}\psi \right)ds, \end{array}$$
where ψ(x,t) is the surface phase field variable, not necessarily the same as ϕ(x,t) confined to the surface, $e_{s}$ is the surface energy density per unit area including the kinetic energy on the surface $\frac{\rho_{s}}{2}\psi_{t}^{2}$ and $\nabla_{s}$ is the surface gradient operator over smooth boundary ∂Ω as those in [45,46].

In the following, we will use the volume and area fraction to illustrate why we use two phase variables or order parameters in bulk and surface, respectively. Actually, the phase field variables can include the mass fraction or density fraction in place of the volume fraction. We define the effective volume of the two particles as $v_{1}$ and $v_{2}$ and assume there are $N_{1}$ and $N_{2}$ particles, respectively, in a material volume $\delta V=N_{1}v_{1}+N_{2}v_{2}$. Analogously, we define the effective areas of the two particles’ cross-section in the surface as $s_{1}$ and $s_{2}$ and assume there are $N_{1}^{s}$ and $N_{2}^{s}$ particles in a material surface $\delta S=N_{1}^{s}s_{1}+N_{2}^{s}s_{2}$. Then, the bulk volume fraction ϕ and the surface area fraction ψ for the first material can be defined as follows:(11)$$\begin{array}{c} \phi \left(x\right)=\frac{N_{1}\left(x\right)v_{1}}{N_{1}\left(x\right)v_{1}+N_{2}\left(x\right)v_{2}}=\frac{N_{1}\left(x\right)}{N_{1}\left(x\right)+N_{2}\left(x\right)\frac{v_{2}}{v_{1}}}, \end{array}$$(12)$$\begin{array}{c} \psi \left(s\right)=\frac{N_{1}^{s}\left(s\right)s_{1}}{N_{1}^{s}\left(s\right)s_{1}+N_{2}^{s}\left(s\right)s_{2}}=\frac{N_{1}^{s}\left(s\right)}{N_{1}^{s}\left(s\right)+N_{2}^{s}\left(s\right)\frac{s_{2}}{s_{1}}}. \end{array}$$
Clearly, there are no reasons to believe $N_{i}=N_{i}^{s}$ nor $\frac{v_{2}}{v_{1}}=\frac{s_{2}}{s_{1}}$. For example, we can choose the volume fraction of the first material component as the order parameter in the bulk while choosing the area fraction of the second material component as the order parameter at the boundary. So, ${\phi |}_{\partial \Omega }$ and ψ can be two different phase field variables. So we introduce coupling energy in the model to account for the free energy due to the discrepancy between the two physical quantities
(13)$$\begin{array}{c} E_{c}\left[\phi ,\psi \right]=\int_{\partial \Omega }e_{c}\left(\phi ,\nabla_{s}\phi ,\nabla_{s}\nabla_{s}\phi ,\psi ,\nabla_{s}\psi ,\nabla_{s}\nabla_{s}\psi \right)ds, \end{array}$$
where ${\phi =\phi \left(x,t\right)|}_{\partial \Omega }$ and $e_{c}$ is the coupling energy density per unit area. For example, the coupling free energy can be used to describe different boundary surface properties, such as attractive or repulsive, between the material components in different regions of the boundary [29].

The total free energy of the system is given by
(14)$$\begin{array}{c} \begin{array}{c} E\left[\phi ,\psi \right]=\int_{\Omega }e_{b}\left(\phi ,\phi_{t},\nabla \phi ,\nabla \nabla \phi \right)dx+\int_{\partial \Omega }[e_{s}\left(\psi ,\psi_{t},\nabla_{s}\psi ,\nabla_{s}\nabla_{s}\psi \right)\\ \;\;+e_{c}\left(\phi ,\nabla_{s}\phi ,\nabla_{s}\nabla_{s}\phi ,\psi ,\nabla_{s}\psi ,\nabla_{s}\nabla_{s}\psi \right)]ds. \end{array} \end{array}$$

We calculate the time rate of change of the free energy as follows, assuming domain Ω is fixed,
(15)$$\begin{array}{c} \begin{array}{c} \frac{dE}{dt}=\int_{\Omega }\mu_{b}\phi_{t}dx+\int_{\partial \Omega }[\frac{\partial e_{s}}{\partial \psi_{t}}\psi_{tt}+\frac{\partial (e_{s}+e_{c})}{\partial \psi }\psi_{t}+\frac{\partial (e_{s}+e_{c})}{\partial \nabla_{s}\psi }\nabla_{s}\psi_{t}+\frac{\partial (e_{s}+e_{c})}{\partial \nabla_{s}\nabla_{s}\psi }\nabla_{s}\nabla_{s}\psi_{t}\\ \;+\frac{\partial (e_{s}+e_{c})}{\partial \phi }\phi_{t}+\frac{\partial (e_{s}+e_{c})}{\partial \nabla_{s}\phi }\nabla_{s}\phi_{t}+\frac{\partial (e_{s}+e_{c})}{\partial \nabla_{s}\nabla_{s}\phi }\nabla_{s}\nabla_{s}\phi_{t}+n\cdot \frac{\partial e_{b}}{\partial \nabla \phi }\phi_{t}+\frac{\partial e_{b}}{\partial \nabla \nabla \phi }:n\nabla \phi_{t}-n\nabla :\frac{\partial e_{b}}{\partial \nabla \nabla \phi }\phi_{t}]ds\\ =\int_{\Omega }\mu_{b}\phi_{t}dx+\int_{\partial \Omega }[\rho_{s}\psi_{t}\psi_{tt}+\frac{\partial (e_{s}+e_{c})}{\partial \psi }\psi_{t}-\nabla_{s}\cdot \frac{\partial (e_{s}+e_{c})}{\partial \nabla_{s}\psi }\psi_{t}-2Hn\cdot \frac{\partial (e_{s}+e_{c})}{\partial \nabla_{s}\psi }\psi_{t}-\nabla_{s}\cdot \frac{\partial (e_{s}+e_{c})}{\partial \nabla_{s}\nabla_{s}\psi }\cdot \nabla_{s}\psi_{t}\\ \;-2Hn\cdot \frac{\partial (e_{s}+e_{c})}{\partial \nabla_{s}\nabla_{s}\psi }\cdot \nabla_{s}\psi_{t}+\frac{\partial (e_{s}+e_{c})}{\partial \phi }\phi_{t}-\nabla_{s}\cdot \frac{\partial (e_{s}+e_{c})}{\partial \nabla_{s}\phi }\phi_{t}-2Hn\cdot \frac{\partial (e_{s}+e_{c})}{\partial \nabla_{s}\phi }\phi_{t}-\nabla_{s}\cdot \frac{\partial (e_{s}+e_{c})}{\partial \nabla_{s}\nabla_{s}\phi }\cdot \nabla_{s}\phi_{t}\\ \;-2Hn\cdot \frac{\partial (e_{s}+e_{c})}{\partial \nabla_{s}\nabla_{s}\phi }\cdot \nabla_{s}\phi_{t}+n\cdot \frac{\partial e_{b}}{\partial \nabla \phi }\phi_{t}+\frac{\partial e_{b}}{\partial \nabla \nabla \phi }:n\nabla \phi_{t}-\left(n\nabla :\frac{\partial e_{b}}{\partial \nabla \nabla \phi }\right)\phi_{t}]ds\\ =\int_{\Omega }\mu_{b}\phi_{t}dx+\int_{\partial \Omega }\left[\mu_{s}\psi_{t}+\mu_{c}\phi_{t}+\mu_{g}\left(n\cdot \nabla \phi_{t}\right)\right]ds, \end{array} \end{array}$$
where *H* is the mean curvature of the boundary, n is the unit external normal of ∂Ω, the bulk chemical potential $\mu_{b}$, surface chemical potentials $\mu_{s}$, $\mu_{c}$, and the conjugate variable $\mu_{g}$ are given, respectively, by
(16)$$\begin{array}{c} \begin{array}{c} \mu_{b}=\rho \phi_{tt}+\frac{\partial e_{b}}{\partial \phi }-\nabla \cdot \frac{\partial e_{b}}{\partial \nabla \phi }+\nabla \nabla :\frac{\partial e_{b}}{\partial \nabla \nabla \phi },\\ \mu_{s}=\rho_{s}\psi_{tt}+\frac{\partial (e_{s}+e_{c})}{\partial \psi }-\nabla_{s}\cdot \frac{\partial (e_{s}+e_{c})}{\partial \nabla_{s}\psi }-2Hn\cdot \frac{\partial (e_{s}+e_{c})}{\partial \nabla_{s}\psi }+\nabla_{s}\nabla_{s}:\frac{\partial (e_{s}+e_{c})}{\partial \nabla_{s}\nabla_{s}\psi }+2Hn\nabla_{s}:\frac{\partial (e_{s}+e_{c})}{\partial \nabla_{s}\nabla_{s}\psi }+\\ \;\nabla_{s}\cdot \left(2Hn\cdot \frac{\partial (e_{s}+e_{c})}{\partial \nabla_{s}\nabla_{s}\psi }\right)+4H^{2}nn:\frac{\partial (e_{s}+e_{c})}{\partial \nabla_{s}\nabla_{s}\psi },\\ \mu_{c}=\frac{\partial e_{c}}{\partial \phi }-\nabla_{s}\cdot \frac{\partial e_{c}}{\partial \nabla_{s}\phi }-2Hn\cdot \frac{\partial e_{c}}{\partial \nabla_{s}\phi }+\nabla_{s}\nabla_{s}:\frac{\partial e_{c}}{\partial \nabla_{s}\nabla_{s}\phi }+2Hn\nabla_{s}:\frac{\partial e_{c}}{\partial \nabla_{s}\nabla_{s}\phi }+\\ \;\nabla_{s}\cdot \left(2Hn\cdot \frac{\partial e_{c}}{\partial \nabla_{s}\nabla_{s}\phi }\right)+4H^{2}nn:\frac{\partial e_{c}}{\partial \nabla_{s}\nabla_{s}\phi }+n\cdot \frac{\partial e_{b}}{\partial \nabla \phi }-n\nabla :\frac{\partial e_{b}}{\nabla \nabla \phi }-\nabla_{s}n:\frac{\partial e_{b}}{\partial \nabla \nabla \phi }-2Hnn:\frac{\partial e_{b}}{\partial \nabla \nabla \phi },\\ \mu_{g}=\frac{\partial e_{b}}{\partial \nabla \nabla \phi }:nn. \end{array} \end{array}$$

**Remark** **1.**
*(i) There exist two surface chemical potentials, one corresponding to ϕ is denoted as $\mu_{c}$, while the other corresponding to ψ is denoted as $\mu_{s}$ at the boundary. The surface chemical potential, $\mu_{c}$, includes contributions from the coupling free energy as well as that from the bulk energy confined to the boundary, while $\mu_{s}$ only includes contributions from the surface and coupling energies. (ii) The mean curvature shows up in the surface chemical potentials, indicating that curvature of the boundary affects the surface dynamics. (iii) More surface terms can appear if the free energy density function depends on higher order spatial derivatives. We adopt the Einstein notation for tensors here, denote tensor product of vector n and v as $nv=n_{i}v_{j}$, use one dot · to represent inner product $n\cdot v=n_{i}v_{i}$ and two dots: to represent contraction of two second order tensor $A:C=A_{ij}C_{ij}$, where n,v are vectors, A,C are second order tensors.*


#### 2.2.1. Dynamics in the Bulk

We apply the generalized Onsager principle firstly to the bulk integral in (Equation 15) to obtain the transport equation for ϕ in the bulk, Ω,
(17)$$\begin{array}{c} -M_{b}^{-1}\phi_{t}=\mu_{b}\Leftrightarrow \phi_{t}=-M_{b}\mu_{b},\;x\in \Omega , \end{array}$$
where $M_{b}$ is the mobility operator and $M_{b}^{-1}$ is the friction operator which is positive semi-definite to ensure energy dissipation. We limit the mobility operator in the following form in this study
(18)$$\begin{array}{c} M_{b}=M_{b}^{\left(1\right)}-\nabla \cdot M_{b}^{\left(2\right)}\cdot \nabla , \end{array}$$
where $M_{b}^{\left(1\right)}\geq 0$ is a scalar function of ϕ and $M_{b}^{\left(2\right)}\in R^{3\times 3}$ is a semi-definite positive matrix which can be a function of ϕ as well. If $M_{b}^{\left(2\right)}=M_{b}^{\left(2\right)}I$ and $M_{b}^{\left(2\right)}$ is also a scalar function of ϕ, then a special case $\nabla \cdot M_{b}^{\left(2\right)}\cdot \nabla =\nabla \cdot \left(M_{b}^{\left(2\right)}\nabla \right)$ can be obtained. We note that the derivation applies to a more general mobility operator with high order derivatives as well, which we will not pursue in this study. The presence of spatial derivatives in the mobility indicates the non-local interaction is accounted for in friction operator $M_{b}^{-1}$. This is shown in the form of pseudo-differential operators. With this, the energy dissipation rate reduces to
(19)$$\begin{array}{c} \begin{array}{c} \frac{dE}{dt}=-\int_{\Omega }\left[\mu_{b}M_{b}^{\left(1\right)}\mu_{b}+\nabla \mu_{b}\cdot M_{b}^{\left(2\right)}\cdot \nabla \mu_{b}\right]dx+\int_{\partial \Omega }\left[\mu_{s}\psi_{t}+\mu_{c}\phi_{t}+\mu_{g}\nabla_{n}\phi_{t}+\mu_{b}n\cdot M_{b}^{\left(2\right)}\cdot \nabla \mu_{b}\right]ds. \end{array} \end{array}$$

We remark that $n\cdot M_{b}^{\left(2\right)}\cdot \nabla \mu_{b}$ is the flux of the volume fraction across the boundary. This physical quantity is determined by the balance between the surface and bulk chemical potential. We next derive consistent boundary conditions for this model.

#### 2.2.2. Dynamics on the Boundary

We recognize that the boundary energy flux density is a quadratic form and then apply the Onsager principle the second time to the energy flux density to establish a dynamical constitutive equation at the boundary:(20)$$\left(\begin{array}{c} \phi_{t}\\ \psi_{t}\\ f_{m}\\ \nabla_{n}\phi_{t} \end{array}\right)=-M_{4\times 4}\cdot \left(\begin{array}{c} \mu_{c}\\ \mu_{s}\\ \mu_{b}\\ \mu_{g} \end{array}\right),$$
where $f_{m}=n\cdot M_{b}^{\left(2\right)}\cdot \nabla \mu_{b}$ is the inward volume flux, $\phi_{t},\psi_{t},\nabla_{n}\phi_{t}$ are time rate of changes of three quantities, identified as generalized fluxes, and $M_{4\times 4}\geq 0$ is the surface mobility operator, a 4×4 matrix or second-order tensor. $M_{4\times 4}\geq 0$ means that its symmetric part is semi-positive definite. Then,
(21)$$\begin{array}{c} \begin{array}{c} \frac{dE}{dt}=-\int_{\Omega }\left[\mu_{b}M_{b}^{\left(1\right)}\mu_{b}+\nabla \mu_{b}\cdot M_{b}^{\left(2\right)}\cdot \nabla \mu_{b}\right]dx-\int_{\partial \Omega }\left[\left(\mu_{c},\mu_{s},\mu_{b},\mu_{g}\right)\left(M_{4\times 4}\right){\left(\mu_{c},\mu_{s},\mu_{b},\mu_{g}\right)}^{T}\right]ds\leq 0, \end{array} \end{array}$$
which indicates the system is dissipative. The linear response relation in (Equation 20) links the generalized fluxes with the generalized forces which gives a very general boundary surface dynamical system. We next give two examples of the mobility operator to make contact with the existing thermodynamically consistent phase field models in the literature.

**Example** **1**
*(First type dynamic boundary conditions (DBCs): a purely dissipative boundary condition). We specify a symmetric mobility operator as follows*

(22)
$$\begin{array}{c} M_{4\times 4}=\left(\begin{array}{cccc} M_{c}+\frac{\gamma^{2}}{\alpha } & \frac{\beta \gamma }{\alpha } & -\frac{\gamma }{\alpha } & \frac{\gamma \theta }{\alpha }\\ \frac{\beta \gamma }{\alpha } & M_{s}+\frac{\beta^{2}}{\alpha } & -\frac{\beta }{\alpha } & \frac{\beta \theta }{\alpha }\\ -\frac{\gamma }{\alpha } & -\frac{\beta }{\alpha } & \frac{1}{\alpha } & -\frac{\theta }{\alpha }\\ \frac{\gamma \theta }{\alpha } & \frac{\beta \theta }{\alpha } & -\frac{\theta }{\alpha } & M_{g}+\frac{\theta^{2}}{\alpha } \end{array}\right). \end{array}$$

*where $M_{c},M_{s},M_{g}\geq 0$ are three semi-definite positive operators, α≥0 is a friction coefficient, β,γ,θ are weight coefficients which can take arbitrary values.*

*This constitutive equation establishes a balance between the inward volume flux at the boundary and the generalized chemical potential difference between the bulk and the surface: it assumes the inward volume flux is proportional to the difference between the chemical potential in the bulk and the weighted one at the boundary. When the weighted surface energy is higher than the bulk energy confined to the boundary, the volume flux is inward; otherwise, the volume flux flows outward. In either case, the total energy dissipates since $M_{4\times 4}\geq 0$.*

*The governing equation together with the DBCs in this model is summarized as follows*

(23)
$$\begin{array}{c} \begin{array}{c} \phi_{t}=-M_{b}\mu_{b},\;x\in \Omega ,\\ \phi_{t}=-\left(M_{c}+\frac{\gamma^{2}}{\alpha }\right)\mu_{c}-\frac{\beta \gamma }{\alpha }\mu_{s}+\frac{\gamma }{\alpha }\mu_{b}-\frac{\gamma \theta }{\alpha }\mu_{g},\;\psi_{t}=-\frac{\beta \gamma }{\alpha }\mu_{c}-\left(M_{s}+\frac{\beta^{2}}{\alpha }\right)\mu_{s}+\frac{\beta }{\alpha }\mu_{b}-\frac{\beta \theta }{\alpha }\mu_{g},\;s\in \partial \Omega ,\\ \alpha n\cdot M_{b}^{\left(2\right)}\cdot \nabla \mu_{b}=-\mu_{b}+\beta \mu_{s}+\gamma \mu_{c}+\theta \mu_{g},\;\nabla_{n}\phi_{t}=-\frac{\gamma \theta }{\alpha }\mu_{c}-\frac{\beta \theta }{\alpha }\mu_{s}+\frac{\theta }{\alpha }\mu_{b}-\left(M_{g}+\frac{\theta^{2}}{\alpha }\right)\mu_{g},\;s\in \partial \Omega . \end{array} \end{array}$$


*The corresponding energy dissipation rate is given by*

(24)
$$\begin{array}{c} \begin{array}{c} \frac{dE}{dt}=-\int_{\Omega }\left[\mu_{b}M_{b}^{\left(1\right)}\mu_{b}+\nabla \mu_{b}\cdot M_{b}^{\left(2\right)}\cdot \nabla \mu_{b}\right]dx-\int_{\partial \Omega }\mu_{g}M_{g}\mu_{g}\;ds\\ -\int_{\partial \Omega }\mu_{c}M_{c}\mu_{c}ds-\int_{\partial \Omega }\mu_{s}M_{s}\mu_{s}ds-\frac{1}{\alpha }\int_{\partial \Omega }{\left(\gamma \mu_{c}+\beta \mu_{s}+\theta \mu_{g}-\mu_{b}\right)}^{2}ds. \end{array} \end{array}$$

*The surface transport equation of ϕ can be rewritten into an alternative and more suggestive form on boundary ∂Ω as follows*

(25)
$$\begin{array}{c} \begin{array}{c} \phi_{t}=-M_{c}\mu_{c}-\gamma f_{m},\;\psi_{t}=-M_{s}\mu_{s}-\beta f_{m},\;\frac{\partial \nabla_{n}\phi }{\partial t}=-M_{g}\mu_{g}-\theta f_{m},\;s\in \partial \Omega , \end{array} \end{array}$$

*where the inward volume fraction flux is stipulated as follows at the boundary*

(26)
$$\begin{array}{c} f_{m}=n\cdot M_{b}^{\left(2\right)}\cdot \nabla \mu_{b}=\frac{1}{\alpha }\left(\beta \mu_{s}+\gamma \mu_{c}+\theta \mu_{g}-\mu_{b}\right). \end{array}$$


*Notice that β,γ,θ can be of any numerical values and if $\mu_{g}=0$, we set θ=0 and drop $n\cdot \nabla \phi_{t}$ from the constitutive equation in (Equation 20). (Equation 25) indicates that so long as the volume fraction flux is balanced by (Equation 26), the relaxation dynamics of the three variables ϕ,ψ,n·∇ϕ at the boundary are dictated by the differences between the corresponding chemical potentials and the weighted inward volume fraction flux, where the weights are delicately tied to the volume fraction flux at the boundary. This gives one a great deal of flexibility to fine-tune the volume fraction flux at the boundary to control the bulk dynamics. So, there is no surprise that this model includes many existing thermodynamically consistent phase field models with DBCs in the literature [23,24,27,28,29].*


**Example** **2**
*(Second DBCs: a dissipative and transportive boundary condition). In the second example, we specify the mobility operator in another form with a non-zero antisymmetric component:*

(27)
$$M_{4\times 4}=\left(\begin{array}{cccc} M_{c} & 0 & \frac{\gamma }{\alpha } & 0\\ 0 & M_{s} & \frac{\beta }{\alpha } & 0\\ -\frac{\gamma }{\alpha } & -\frac{\beta }{\alpha } & \frac{1}{\alpha } & -\frac{\theta }{\alpha }\\ 0 & 0 & \frac{\theta }{\alpha } & M_{g} \end{array}\right)=\left(\begin{array}{cccc} M_{c} & 0 & 0 & 0\\ 0 & M_{s} & 0 & 0\\ 0 & 0 & \frac{1}{\alpha } & 0\\ 0 & 0 & 0 & M_{g} \end{array}\right)+\left(\begin{array}{ccccc} 0 & 0 & \frac{\gamma }{\alpha } & 0 & \\ 0 & 0 & \frac{\beta }{\alpha } & 0\\ -\frac{\gamma }{\alpha } & -\frac{\beta }{\alpha } & 0 & -\frac{\theta }{\alpha }\\ 0 & 0 & \frac{\theta }{\alpha } & 0 \end{array}\right),$$

*where the antisymmetric component contributes to transport dynamics at the boundary, in addition to the dissipative part given by the positive semi-definite operator in $M_{4\times 4}$. The antisymmetric mobility component represents an energy exchange between the bulk and the boundary without inducing any dissipation.*

*The governing system of equations together with the dynamic boundary conditions in this model is summarized as follows*

(28)
$$\begin{array}{c} \begin{array}{c} \phi_{t}=-M_{b}\mu_{b},\;x\in \Omega ,\\ \phi_{t}=-M_{c}\mu_{c}-\frac{\gamma }{\alpha }\mu_{b},\;\psi_{t}=-M_{s}\mu_{s}-\frac{\beta }{\alpha }\mu_{b},\;s\in \partial \Omega ,\\ \alpha n\cdot M_{b}^{\left(2\right)}\cdot \nabla \mu_{b}=-\mu_{b}+\beta \mu_{s}+\gamma \mu_{c}+\theta \mu_{g},\;\nabla_{n}\phi_{t}=-M_{g}\mu_{g}-\frac{\theta }{\alpha }\mu_{b},\;s\in \partial \Omega . \end{array} \end{array}$$

*The corresponding energy dissipation rate is given by*

(29)
$$\begin{array}{c} \begin{array}{c} \frac{dE}{dt}=-\int_{\Omega }\left[\mu_{b}M_{b}^{\left(1\right)}\mu_{b}+\nabla \mu_{b}\cdot M_{b}^{\left(2\right)}\cdot \nabla \mu_{b}\right]dx-\int_{\partial \Omega }\left[\mu_{c}M_{c}\mu_{c}+\mu_{s}M_{s}\mu_{s}+\frac{1}{\alpha }\mu_{b}^{2}+\mu_{g}M_{g}\mu_{g}\right]ds. \end{array} \end{array}$$


*Notice that the mobility matrix has an antisymmetric component that does not contribute to the energy dissipation. In fact, if we change the antisymmetric part into a symmetric one by negating signs of the non-zero off-diagonal entries in the antisymmetric operator, we recover the model given in the previous example. This set of boundary conditions has the following interpretation: the time rate of changes of the variables at the boundary is proportional to the differences between the surface chemical potentials and weighted bulk chemical potentials. The dynamics are thus different from the previous example and the energy dissipation rate is altered as well.*


The two examples of dynamic boundary conditions are derived from two different considerations of mobility operators, as well as effective chemical potentials, which contribute to distinct energy dissipation mechanisms, following the generalized Onsager principle under a unified assumption that the boundary volume fraction flux is proportional to the difference of the bulk chemical potential confined at the boundary and weighted surface energy. In the first case, the time rate of change of the volume fraction is proportional to the difference between the surface chemical potential and the outward volume flux. As a result, the distinctive surface energy dissipation rate is directly linked to the magnitude of the volume fraction flux across the boundary surface. In the second case, the time rate of change of the volume fraction is proportional to the difference between the surface chemical potential and the weighted bulk chemical potential confined to the boundary. Consequentially, the distinctive surface energy dissipation rate is measured by the bulk chemical potential confined to the surface, and the other surface chemical potentials. Two different dissipative mechanisms define two different dynamical models at the boundary. There are more cases that one can elaborate by specifying specific form of operator $M_{4\times 4}$, which we will not enumerate here.

### 2.3. Effect of Mobilities in the Bulk and on the Surface

In general, mobility operator in the bulk $M_{b}=M_{b}^{sym}+M_{b}^{anti}$ in (Equation 17) is decomposed into symmetric and antisymmetric parts, where $M_{b}^{sym}$ is semi-definite positive. There can be many thermodynamically consistent boundary conditions that are compatible to the given bulk transport equation. However, the dissipative property of the boundary transport equation system depends not only on the mobility operator, but also on the geometry of the boundary as well. Now let us examine the special case where $M_{b}^{\left(2\right)}=0$, the energy dissipation rate in (Equation 19) is given by
(30)$$\begin{array}{c} \begin{array}{c} \frac{dE}{dt}=-\int_{\Omega }\mu_{b}M_{b}^{\left(1\right)}\mu_{b}dx+\int_{\partial \Omega }\left[\mu_{s}\psi_{t}+\mu_{c}\phi_{t}+\mu_{g}\nabla_{n}\phi_{t}\right]ds. \end{array} \end{array}$$
One choice of the surface dynamics is given by
(31)$$\left(\begin{array}{c} \phi_{t}\\ \psi_{t}\\ \nabla_{n}\phi_{t} \end{array}\right)=-\left(\begin{array}{ccc} M_{c} & 0 & 0\\ 0 & M_{s} & 0\\ 0 & 0 & M_{g} \end{array}\right)\cdot \left(\begin{array}{c} \mu_{c}\\ \mu_{s}\\ \mu_{g} \end{array}\right),$$
which is the limiting case for (Equation 23) and (Equation 28) with $M_{b}^{\left(2\right)}=0$ and α→+∞. The general models with two types of dynamic boundary conditions reduce to the same set of dynamic boundary conditions. For the more general case, where $M_{b}=\sum_{k=0}^{K}{\left(-1\right)}^{k}M_{b}^{\left(k\right)}\nabla^{2k}$,
(32)$$\begin{array}{c} \begin{array}{c} \frac{dE}{dt}=-\int_{\Omega }\sum_{k=0}^{K}\nabla^{k}\mu_{b}M_{b}^{\left(k\right)}\nabla^{k}\mu_{b}dx+\int_{\partial \Omega }[\mu_{s}\psi_{t}+\mu_{c}\phi_{t}+\mu_{g}\nabla_{n}\phi_{t}\\ \;\;+\sum_{k=0}^{k}{\left(-1\right)}^{k}M_{b}^{\left(k\right)}\sum_{i=0}^{K}{\left(-1\right)}^{i}\nabla^{i}\mu_{b}\cdot n\cdot \nabla^{2k-i-1}\mu_{b}]ds. \end{array} \end{array}$$
To make the energy dissipation rate non-positive, suitable boundary conditions for the unknowns must be taken into consideration. However, this is not our focus here.

For the mobility operators at the boundary, we consider the following forms, analogously to the bulk,
(33)$$\begin{array}{c} M_{c}=M_{c}^{\left(1\right)}-\nabla_{s}\cdot M_{c}^{\left(2\right)}\cdot \nabla_{s},\;M_{s}=M_{s}^{\left(1\right)}-\nabla_{s}\cdot M_{s}^{\left(2\right)}\cdot \nabla_{s},\;M_{g}=M_{g}^{\left(1\right)}-\nabla_{s}\cdot M_{g}^{\left(2\right)}\cdot \nabla_{s}, \end{array}$$
where $M_{c}^{\left(1\right)}\geq 0$, $M_{s}^{\left(1\right)}\geq 0$, $M_{g}^{\left(1\right)}\geq 0$, $M_{c}^{\left(2\right)}$, $M_{s}^{\left(2\right)}$ and $M_{g}^{\left(2\right)}$ are 3×3 positive semi-definite matrices. Then,
(34)$$\begin{array}{c} \begin{array}{c} -\int_{\partial \Omega }\left[\mu_{c}M_{c}\mu_{c}\right]ds=-\int_{\partial \Omega }\left[\mu_{c}M_{c}^{\left(1\right)}\mu_{c}+\nabla_{s}\mu_{c}\cdot M_{c}^{\left(2\right)}\cdot \nabla_{s}\mu_{c}+2H\mu_{c}n\cdot M_{c}^{\left(2\right)}\cdot \nabla_{s}\mu_{c}\right]ds,\\ -\int_{\partial \Omega }\left[\mu_{s}M_{s}\mu_{s}\right]ds=-\int_{\partial \Omega }\left[\mu_{s}M_{s}^{\left(1\right)}\mu_{s}+\nabla_{s}\mu_{s}\cdot M_{s}^{\left(2\right)}\cdot \nabla_{s}\mu_{s}+2H\mu_{s}n\cdot M_{s}^{\left(2\right)}\cdot \nabla_{s}\mu_{s}\right]ds,\\ -\int_{\partial \Omega }\left[\mu_{g}M_{g}\mu_{g}\right]ds=-\int_{\partial \Omega }\left[\mu_{g}M_{g}^{\left(1\right)}\mu_{g}+\nabla_{s}\mu_{g}\cdot M_{g}^{\left(2\right)}\cdot \nabla_{s}\mu_{g}+2H\mu_{g}n\cdot M_{g}^{\left(2\right)}\cdot \nabla_{s}\mu_{g}\right]ds. \end{array} \end{array}$$
Whether or not the energy dissipation rate at the boundary is non-positive depends on the last term in all three lines in (34), which are linearly proportional to the mean curvature, surface chemical potential, and the corresponding surface flux of each variable.

Having discussed the energy dissipative property of the model, let us investigate how the volume fraction variable evolves dynamically by examining special cases. If $M_{c}^{\left(2\right)}=M_{c}^{\left(2\right)}I$, $M_{s}^{\left(2\right)}=M_{s}^{\left(2\right)}I$ and $M_{g}^{\left(2\right)}=M_{g}^{\left(2\right)}I$, where $M_{c}^{\left(2\right)}\geq 0$, $M_{s}^{\left(2\right)}\geq 0$ and $M_{g}^{\left(2\right)}\geq 0$ are scalar functions of ϕ, the last term in each line of (34) vanishes and the energy dissipation rate at the boundary is non-positive due to $n\cdot \nabla_{s}\mu_{c}=n\cdot \nabla_{s}\mu_{s}=n\cdot \nabla_{s}\mu_{g}=0$. Of course, H=0 is also a sufficient condition for non-positive energy dissipation rates.

It follows from (Equation 23) that
(35)$$\begin{array}{c} \begin{array}{c} \frac{d}{dt}\left[\int_{\Omega }\beta \phi dx+\int_{\partial \Omega }\psi ds\right]=-\beta \int_{\Omega }M_{b}^{\left(1\right)}\mu_{b}dx-\int_{\partial \Omega }\left[M_{s}^{\left(1\right)}\mu_{s}+2Hn\cdot M_{s}^{\left(2\right)}\cdot \nabla_{s}\mu_{s}\right]ds, \end{array} \end{array}$$

If $M_{b}^{\left(1\right)}=M_{s}^{\left(1\right)}=0$ and $M_{s}^{\left(2\right)}=M_{s}^{\left(2\right)}I$ or H=0,
(36)$$\begin{array}{c} \frac{d}{dt}\left[\int_{\Omega }\beta \phi dx+\int_{\partial \Omega }\psi ds\right]=0. \end{array}$$
This indicates a β−weighted volume fraction in the bulk and the area fraction over the surface is conserved under this dynamic boundary condition within the Cahn–Hilliard surface dynamics. Since β is not limited to be positive, this result simply reveals a delicate balance between the volume fraction in the bulk and that on the surface must be maintained during the dynamical process at all times.

Similarly, it also follows from (Equation 23) that
(37)$$\begin{array}{c} \begin{array}{c} \frac{d}{dt}\left[\int_{\Omega }\gamma \phi dx+\int_{\partial \Omega }\phi ds\right]=-\gamma \int_{\Omega }M_{b}^{\left(1\right)}\mu_{b}dx-\int_{\partial \Omega }\left[M_{c}^{\left(1\right)}\mu_{c}+2Hn\cdot M_{c}^{\left(2\right)}\cdot \nabla_{s}\mu_{c}\right]ds, \end{array} \end{array}$$

If $M_{b}^{\left(1\right)}=M_{c}^{\left(1\right)}=0$ and $M_{c}^{\left(2\right)}=M_{c}^{\left(2\right)}I$ or H=0,
(38)$$\begin{array}{c} \frac{d}{dt}\left[\int_{\Omega }\gamma \phi dx+\int_{\partial \Omega }\phi ds\right]=0. \end{array}$$
This replicates an analogous delicate balance between the bulk volume fraction and the surface one that is preserved in the dynamical process in the Cahn–Hilliard surface dynamics, where γ is that delicate weight parameter with potentially an arbitrary value.

Model (Equation 23) and (Equation 28) give two fairly general phase field models with two different dynamic boundary conditions, where the surface transport equations of phase variables at the boundary set the two models apart. In the first one, the cross-boundary volume fraction flux contributes directly to the energy dissipation on the surface; while in the second, it is the bulk chemical potential limited to the boundary that contributes to the energy dissipation on the surface directly. We next examine various limiting cases (Equation 23) and (Equation 28), respectively, and show that they reduce to the thermodynamically consistent phase field models in the literature. In fact, when the cross-boundary volume fraction flux (inward or outward flux) vanishes, i.e., α=∞ and γ=β=θ=0, the two types of dynamic boundary conditions are identical.

## 3. Reduction to Limiting Cases

We examine three limiting cases of the model to make contact with the existing thermodynamically consistent phase field models and then present a new reactive transport limiting model with reactive dynamic boundary conditions.

### 3.1. The Jing–Wang Model [24]

We choose $e_{c}=\frac{1}{2K}{\left(\phi -\psi \right)}^{2}$, where K>0 is a penalizing parameter, and define a symmetric mobility operator as follows
(39)$$\begin{array}{c} M_{4\times 4}=\left(\begin{array}{cccc} M_{s}+\frac{\beta^{2}}{\alpha } & M_{s}+\frac{\beta^{2}}{\alpha } & -\frac{\beta }{\alpha } & \frac{\beta \theta }{\alpha }\\ M_{s}+\frac{\beta^{2}}{\alpha } & M_{s}+\frac{\beta^{2}}{\alpha } & -\frac{\beta }{\alpha } & \frac{\beta \theta }{\alpha }\\ -\frac{\beta }{\alpha } & -\frac{\beta }{\alpha } & \frac{1}{\alpha } & -\frac{\theta }{\alpha }\\ \frac{\beta \theta }{\alpha } & \frac{\beta \theta }{\alpha } & -\frac{\theta }{\alpha } & M_{g} \end{array}\right). \end{array}$$

The governing equation system together with the boundary conditions in this model is given as follows
(40)$$\begin{array}{c} \begin{array}{c} \phi_{t}=-M_{b}\mu_{b},\;x\in \Omega ,\\ \psi_{t}=-\left(M_{s}+\frac{\beta^{2}}{\alpha }\right)\left(\mu_{s}+\mu_{c}\right)+\frac{\beta }{\alpha }\mu_{b}-\frac{\beta \theta }{\alpha }\mu_{g},\;\phi_{t}=-\left(M_{s}+\frac{\beta^{2}}{\alpha }\right)\left(\mu_{s}+\mu_{c}\right)+\frac{\beta }{\alpha }\mu_{b}-\frac{\beta \theta }{\alpha }\mu_{g},\;s\in \partial \Omega ,\\ \alpha n\cdot M_{b}^{\left(2\right)}\cdot \nabla \mu_{b}=-\mu_{b}+\beta \left(\mu_{s}+\mu_{c}\right)+\theta \mu_{g},\;\nabla_{n}\phi_{t}=-M_{g}\mu_{g}-\theta f_{m},\;s\in \partial \Omega . \end{array} \end{array}$$
When K→0, ${\psi =\phi |}_{\partial \Omega }$ and the governing equation system reduces to
(41)$$\begin{array}{c} \begin{array}{c} \phi_{t}=-M_{b}\mu_{b},\;x\in \Omega ,\\ \phi_{t}=-\left(M_{s}+\frac{\beta^{2}}{\alpha }\right)\left(\mu_{s}+\mu_{c}\right)+\frac{\beta }{\alpha }\mu_{b}-\frac{\beta \theta }{\alpha }\mu_{g},\;s\in \partial \Omega ,\\ \alpha n\cdot M_{b}^{\left(2\right)}\cdot \nabla \mu_{b}=-\mu_{b}+\beta \left(\mu_{s}+\mu_{c}\right)+\theta \mu_{g},\;\nabla_{n}\phi_{t}=-M_{g}\mu_{g}-\theta f_{m},\;s\in \partial \Omega . \end{array} \end{array}$$
The corresponding energy dissipation rate is given by
(42)$$\begin{array}{c} \begin{array}{c} \frac{dE}{dt}=-\int_{\Omega }\left[\mu_{b}M_{b}^{\left(1\right)}\mu_{b}+\nabla \mu_{b}\cdot M_{b}^{\left(2\right)}\cdot \nabla \mu_{b}\right]dx-\int_{\partial \Omega }\left[\mu_{g}M_{g}\mu_{g}\right]ds-\\ \int_{\partial \Omega }\left[\left(\mu_{c}+\mu_{s}\right)M_{s}\left(\mu_{c}+\mu_{s}\right)\right]ds-\frac{1}{\alpha }\int_{\partial \Omega }{\left(\beta \left(\mu_{c}+\mu_{s}\right)+\theta \mu_{g}-\mu_{b}\right)}^{2}ds. \end{array} \end{array}$$
Setting θ=0, this system reduces to the general model with the first type dynamic boundary condition derived in [24] by the authors. Similarly, we deduce the general model with the second type dynamic boundary condition in [24] by selecting the mobility operator corresponding to the second type boundary condition alluded to in the previous section.

### 3.2. The Knopf–Lam Model [28] and the Liu–Wu Model [27]

We define the free energy densities as follows
(43)$$\begin{array}{c} \begin{array}{c} e_{b}=\frac{\epsilon }{2}{\left|\nabla \phi \right|}^{2}+\frac{1}{\epsilon }F\left(\phi \right),\;e_{s}=\frac{\sigma }{2}{\left|\nabla_{s}\psi \right|}^{2}+\frac{1}{\sigma }G\left(\psi \right),\;e_{c}=\frac{1}{2K}{\left(H\left(\psi \right)-\phi \right)}^{2}. \end{array} \end{array}$$
The corresponding energy dissipation rate is given by
(44)$$\begin{array}{c} \begin{array}{c} \frac{dE}{dt}=\int_{\Omega }\left(-\epsilon \nabla^{2}\phi +\frac{1}{\epsilon }F^{\prime }\left(\phi \right)\right)\phi_{t}dx+\int_{\partial \Omega }\left(\epsilon n\cdot \nabla \phi -\frac{1}{K}\left(H\left(\psi \right)-\phi \right)\right)\phi_{t}ds\\ \;+\int_{\partial \Omega }\left(-\sigma \nabla_{s}^{2}\psi +\frac{1}{\sigma }G^{\prime }\left(\psi \right)+\frac{1}{K}\left(H\left(\psi \right)-\phi \right)H^{\prime }\left(\psi \right)\right)\psi_{t}ds\\ \;=\int_{\Omega }\mu_{b}\phi_{t}dx+\int_{\partial \Omega }\left[\mu_{c}\phi_{t}+\mu_{s}\psi_{t}\right]ds\\ \;=-\int_{\Omega }{\left|\nabla \mu_{b}\right|}^{2}dx+\int_{\partial \Omega }\left[\mu_{c}\phi_{t}+\mu_{s}\psi_{t}+\mu_{b}n\cdot \nabla \mu_{b}\right]ds. \end{array} \end{array}$$
The transport equation in the bulk is given by
(45)$$\begin{array}{c} \begin{array}{c} \phi_{t}=\nabla^{2}\mu_{b},\;\mu_{b}=-\epsilon \nabla^{2}\phi +\frac{1}{\epsilon }F^{\prime }\left(\phi \right),\;x\in \Omega . \end{array} \end{array}$$
We choose such a relationship between the general fluxes and general forces at the surface as follows
(46)$$\left(\begin{array}{c} \phi_{t}\\ \psi_{t}\\ n\cdot \nabla \mu_{b} \end{array}\right)=-\left(\begin{array}{ccc} \kappa_{1} & 0 & 0\\ 0 & -\nabla_{s}^{2} & 0\\ 0 & 0 & \kappa_{2} \end{array}\right)\cdot \left(\begin{array}{c} \mu_{c}\\ \mu_{s}\\ \mu_{b} \end{array}\right),$$
where $\mu_{c}=\epsilon n\cdot \nabla \phi -\frac{1}{K}\left(H\left(\psi \right)-\phi \right),\mu_{s}=-\sigma \nabla_{s}^{2}\psi +\frac{1}{\sigma }G^{\prime }\left(\psi \right)+\frac{1}{K}\left(H\left(\psi \right)-\phi \right)H^{\prime }\left(\psi \right)$, $\kappa_{1},\kappa_{2}\geq 0$. The boundary equations are
(47)$$\begin{array}{c} \phi_{t}=-\kappa_{1}\mu_{c},\;\psi_{t}=\nabla_{s}^{2}\mu_{s},\;n\cdot \nabla \mu_{b}=-\kappa_{2}\mu_{b},\;s\in \partial \Omega . \end{array}$$
Then the energy dissipation rate is given by
(48)$$\begin{array}{c} \begin{array}{c} \frac{dE}{dt}=\int_{\Omega }\left(-\epsilon \nabla^{2}\phi +\frac{1}{\epsilon }F^{\prime }\left(\phi \right)\right)\phi_{t}dx+\int_{\partial \Omega }\left(-\sigma \nabla_{s}^{2}\psi +\frac{1}{\sigma }G^{\prime }\left(\psi \right)+\epsilon H^{\prime }\left(\psi \right)n\cdot \nabla \phi \right)\psi_{t}ds\\ \;=-\int_{\Omega }|\nabla \mu_{b}{|}^{2}dx-\int_{\partial \Omega }[\kappa_{1}\mu_{c}^{2}+|\nabla_{s}\mu_{s}{|}^{2}+\kappa_{2}\mu_{b}^{2}]ds\leq 0. \end{array} \end{array}$$

In the limit $\kappa_{1}\to \infty ,\kappa_{2}\to 0$, the governing equations in the bulk and on the boundary are given by
(49)$$\begin{array}{c} \begin{array}{c} \phi_{t}=\nabla^{2}\mu_{b},\;\mu_{b}=-\epsilon \nabla^{2}\phi +\frac{1}{\epsilon }F^{\prime }\left(\phi \right),\;x\in \Omega ,\\ \psi_{t}=\nabla_{s}^{2}\mu_{s},\;\mu_{s}=-\sigma \nabla_{s}^{2}\psi +\frac{1}{\sigma }G^{\prime }\left(\psi \right)+\epsilon H^{\prime }\left(\psi \right)n\cdot \nabla \phi ,\;s\in \partial \Omega ,\\ n\cdot \nabla \mu_{b}=0,\;\epsilon n\cdot \nabla \phi =\frac{1}{K}\left(H\left(\psi \right)-\phi \right),\;s\in \partial \Omega . \end{array} \end{array}$$
The general model reduces to the Knopf–Lam model in [28]. Moreover, when H(ψ)=ψ and K→0, H(ψ)→ϕ. The energy dissipation rate in (Equation 44) reduces to
(50)$$\begin{array}{c} \begin{array}{c} \frac{dE}{dt}=\int_{\Omega }\left(-\epsilon \nabla^{2}\phi +\frac{1}{\epsilon }F^{\prime }\left(\phi \right)\right)\phi_{t}dx+\int_{\partial \Omega }\left(\epsilon n\cdot \nabla \phi \right)-\sigma \nabla_{s}^{2}\phi +\frac{1}{\sigma }G^{\prime }\left(\phi \right)\phi_{t}ds\\ =\int_{\Omega }\mu_{b}\phi_{t}dx+\int_{\partial \Omega }\left(\mu_{c}+\mu_{s}\right)\phi_{t}ds. \end{array} \end{array}$$
The thermodynamically consistent model reduces to the Liu–Wu model in [27]
(51)$$\begin{array}{c} \begin{array}{c} \phi_{t}=\nabla^{2}\mu_{b},\;\mu_{b}=-\epsilon \nabla^{2}\phi +\frac{1}{\epsilon }F^{\prime }\left(\phi \right),\;x\in \Omega ,\\ \phi_{t}=\nabla_{s}^{2}\left(\mu_{c}+\mu_{s}\right),\;\mu_{c}+\mu_{s}=\epsilon n\cdot \nabla \phi -\sigma \nabla_{s}^{2}\phi +\frac{1}{\sigma }G^{\prime }\left(\phi \right),\;n\cdot \nabla \mu_{b}=0,\;s\in \partial \Omega . \end{array} \end{array}$$

### 3.3. Non-Local Models including the Knopf–Signori Model [29]

We consider phase field models with a non-local bulk free energy [47,48] given by
(52)$$\begin{array}{c} E_{b}=\int_{\Omega }[\int_{\Omega }\frac{1}{4}J(∥x-{y∥)\left(\phi \left(x,t\right)-\phi \left(y,t\right)\right)}^{2}dy+f\left(\phi \right)]dx, \end{array}$$
where J(∥x∥) is the interaction kernel and *f* is the free energy density for the bulk. This form of free energy is perhaps more generic than the one that depends on spatial derivatives of the phase variable. We call this a strongly non-local free energy. The bulk chemical potential is given by
(53)$$\begin{array}{c} \mu_{b}=\rho \phi_{tt}+\int_{\Omega }J(∥x-y∥)\left(-\phi \left(y,t\right)\right)dy+f^{\prime }\left(\phi \right)+a\left(x\right)\phi \left(x,t\right), \end{array}$$
where $a\left(x\right)=\int_{\Omega }J(∥x-y∥)dy.$ Likewise, we consider the surface energy and coupling energy given, respectively, by
(54)$$\begin{array}{c} E_{s}=\int_{\partial \Omega }[\int_{\partial \Omega }\frac{1}{4}K(∥x-{y∥)\left(\psi \left(x,t\right)-\psi \left(y,t\right)\right)}^{2}ds_{y}+g\left(\psi \right)]ds_{x}, \end{array}$$
(55)$$\begin{array}{c} E_{c}=\int_{\partial \Omega }[\int_{\partial \Omega }\frac{1}{4}L(∥x-{y∥)\left(\phi \left(x,t\right)-\psi \left(y,t\right)\right)}^{2}ds_{y}+h\left(\phi ,\psi \right)]ds_{x}, \end{array}$$
where g(ϕ) is the surface energy density per area and h(ϕ,ψ) is the coupling energy density per area. The surface chemical potentials are obtained as follows
(56)$$\begin{array}{c} \begin{array}{c} \mu_{s}=\rho_{s}\psi_{tt}+\int_{\partial \Omega }K(∥x-y∥)\left(-\psi \left(y,t\right)\right)ds_{y}+g^{\prime }\left(\psi \right)+a_{S}\left(x\right)\psi \left(x,t\right)\\ \;+\frac{1}{2}\int_{\partial \Omega }L(∥x-y∥)\left(-\phi \left(y,t\right)\right)ds_{y}+\frac{1}{2}a_{c}\left(x\right)\psi \left(x,t\right)+\frac{\partial h(\phi ,\psi )}{\partial \psi },\\ \mu_{c}=\frac{1}{2}\int_{\partial \Omega }L(∥x-y∥)\left(-\psi \left(y,t\right)\right)ds_{y}+\frac{1}{2}a_{c}\left(x\right)\phi \left(x,t\right)+\frac{\partial h(\phi ,\psi )}{\partial \phi }, \end{array} \end{array}$$
where $a_{S}\left(x\right)=\int_{\partial \Omega }K(∥x-y∥)ds_{y},a_{c}\left(x\right)=\int_{\partial \Omega }L(∥x-y∥)ds_{y}.$ The total free energy is then given by
(57)$$\begin{array}{c} E=\int_{\Omega }\rho \phi_{t}^{2}/2dx+\int_{\Omega }\rho_{s}\psi_{t}^{2}/2ds+E_{b}+E_{s}+E_{c}. \end{array}$$

We calculate the time rate of change of the free energy as follows
(58)$$\begin{array}{c} \frac{d}{dt}E=\int_{\Omega }\mu_{b}\phi_{t}dx+\int_{\partial \Omega }\mu_{s}\psi_{t}ds_{x}+\int_{\partial \Omega }\mu_{c}\phi_{t}ds_{x}. \end{array}$$
We apply the generalized Onsager principle to the bulk term to arrive at
(59)$$\begin{array}{c} \phi_{t}=-M_{b}\mu_{b},\;x\in \Omega , \end{array}$$
where $M_{b}$ is the mobility operator. For $M_{b}=M_{b}^{\left(1\right)}-\nabla \cdot M_{b}^{\left(2\right)}\cdot \nabla$,
(60)$$\begin{array}{c} \begin{array}{c} \frac{d}{dt}E=-\int_{\Omega }\left[\mu_{b}M_{b}^{\left(1\right)}\mu_{b}+\nabla \mu_{b}\cdot M_{b}^{\left(2\right)}\cdot \nabla \mu_{b}\right]dx+\int_{\partial \Omega }\left[\mu_{s}\psi_{t}+\mu_{b}n\cdot M_{b}^{\left(2\right)}\cdot \nabla \mu_{b}+\mu_{c}\phi_{t}\right]ds_{x}\\ \;=-\int_{\Omega }\left[\mu_{b}M_{b}^{\left(1\right)}\mu_{b}+\nabla \mu_{b}\cdot M_{b}^{\left(2\right)}\cdot \nabla \mu_{b}\right]dx+\int_{\partial \Omega }\left[\mu_{s}\psi_{t}+\mu_{c}\phi_{t}+\mu_{b}f_{m}\right]ds_{x}. \end{array} \end{array}$$
We specify boundary dynamic equations by using the generalized Onsager principle as follows
(61)$$\left(\begin{array}{c} \phi_{t}\\ \psi_{t}\\ f_{m} \end{array}\right)=-M_{3\times 3}\cdot \left(\begin{array}{c} \mu_{c}\\ \mu_{s}\\ \mu_{b} \end{array}\right),\;s\in \partial \Omega ,$$
where $M_{3\times 3}$ is the non-negative definite boundary mobility operator.

The energy dissipation rate is given by
(62)$$\begin{array}{c} \begin{array}{c} \frac{d}{dt}E=-\int_{\Omega }\left[\mu_{b}M_{b}^{\left(1\right)}\mu_{b}+\nabla \mu_{b}\cdot M_{b}^{\left(2\right)}\cdot \nabla \mu_{b}\right]dx-\int_{\partial \Omega }\left[\left(\mu_{c},\mu_{s},\mu_{b}\right)\cdot M_{3\times 3}\cdot {\left(\mu_{c},\mu_{s},\mu_{b}\right)}^{T}\right]ds_{x}. \end{array} \end{array}$$
When setting
(63)$$\begin{array}{c} \begin{array}{c} M_{3\times 3}=\left(\begin{array}{ccc} \kappa_{3} & 0 & 0\\ 0 & -M_{s}^{\left(2\right)}\nabla_{s}^{2}+\frac{\beta^{2}}{\alpha } & -\frac{\beta }{\alpha }\\ 0 & -\frac{\beta }{\alpha } & \frac{1}{\alpha } \end{array}\right), \end{array} \end{array}$$
the general model reduces to the non-local model with dynamic boundary conditions in [29] in the limit, $\kappa_{3}\to \infty$.

### 3.4. Reactive Transport Equation in a Binary Polymeric System

Finally, we present a new phase field model for a binary reactive polymeric system and assume the two polymer chains are long enough so that they can be viewed as ideal chains. We also assume there are reactions in both the bulk and the surface described as follows,
(64)$$\begin{array}{c} A\overset{k_{b}^{+}}{\underset{k_{b}^{-}}{⇄}}B,\;\mathrm{in}\;\Omega , \end{array}$$
(65)$$\begin{array}{c} A\overset{k_{s}^{+}}{\underset{k_{s}^{-}}{⇄}}B,\;\mathrm{in}\;\partial \Omega , \end{array}$$
where $k_{i}^{+},k_{i}^{-},i=b,s$ are the forward and backward reaction rates in the bulk and on the surface, respectively. The reaction rates in the bulk and surface may be different due to different pressure, exposure to catalysts, enzymes, light, etc.

The corresponding bulk free energy of the reactive system is given by
(66)$$\begin{array}{c} \begin{array}{c} E_{b}=\int_{\Omega }\frac{b^{2}}{6}\nabla \phi \cdot \nabla \phi +\frac{1}{n_{1}}\phi \left(\ln\phi -1-\ln Q_{1}\right)+\frac{1}{n_{2}}\left(1-\phi \right)\left(\ln\left(1-\phi \right)-1-\ln Q_{2}\right)+\chi \phi \left(1-\phi \right)dx, \end{array} \end{array}$$
and the surface and coupling free energies are, respectively, given by
(67)$$\begin{array}{c} \begin{array}{c} E_{s}=\int_{\partial \Omega }\frac{b^{2}}{6}\nabla_{s}\phi \cdot \nabla_{s}\phi +\frac{1}{n_{1}}\psi \left(\ln\psi -1-\ln Q_{1}\right)+\frac{1}{n_{2}}\left(1-\psi \right)\left(\ln\left(1-\psi \right)-1-\ln Q_{2}\right)+\chi_{s}\psi \left(1-\psi \right)ds,\\ E_{c}=\int_{\partial \Omega }\frac{\eta }{2}{\left(H\left(\psi \right)-\phi \right)}^{2}ds, \end{array} \end{array}$$
where ϕ,ψ are two order parameters to describe the volume fraction and the area fraction of the polymer segments in the bulk and the surface, respectively, *b* is the scaled Kuhn length of the polymer segment, $Q_{1},Q_{2}$ are the dimensionless partition functions for the polymer segments, $n_{i},i=1,2,$ are the polymerization index for the two polymer chains, and $\chi ,\chi_{s}$ measure the mixing energy due to the volume exclusion effects between segments in the bulk and surface, respectively.

Using the generalized Onsager principle, we obtain the governing equation with the dynamic boundary condition of the first kind:(68)$$\begin{array}{c} \begin{array}{c} \frac{\partial \phi }{\partial t}=-\left(M_{b}^{\left(1\right)}-M_{b}^{\left(2\right)}\nabla^{2}\right)\mu_{b},\;x\in \Omega ,\\ \mu_{b}=-\frac{b^{2}}{3}\nabla^{2}\phi +\frac{1}{n}\left(\ln\phi -\ln Q_{1}\right)-\frac{1}{n}\left(\ln\left(1-\phi \right)-\ln Q_{2}\right)+\chi \left(1-2\phi \right),\;x\in \Omega ,\\ \frac{\partial \phi }{\partial t}=-\left(M_{c}^{\left(1\right)}-M_{c}^{\left(2\right)}\nabla_{s}^{2}\right)\mu_{c}-\gamma M_{b}^{\left(2\right)}n\cdot \nabla \mu_{b},\;\mu_{c}=\frac{b^{2}}{6}n\cdot \nabla \phi -\eta \left(H\left(\psi \right)-\phi \right),\;s\in \partial \Omega ,\\ \frac{\partial \psi }{\partial t}=-\left(M_{s}^{\left(1\right)}-M_{s}^{\left(2\right)}\nabla_{s}^{2}\right)\mu_{s}-\beta M_{b}^{\left(2\right)}n\cdot \nabla \mu_{b},\;s\in \partial \Omega ,\\ \mu_{s}=-\frac{b^{2}}{3}\nabla_{s}^{2}\phi_{s}+\frac{1}{n}\left(\ln\psi -\ln Q_{1}\right)-\frac{1}{n}\left(\ln\left(1-\psi \right)-\ln Q_{2}\right)+\chi_{s}\left(1-2\psi \right)+\eta H^{\prime }\left(\psi \right)\left(H\left(\psi \right)-\phi \right),\;s\in \partial \Omega ,\\ \alpha M_{b}^{\left(2\right)}n\cdot \nabla \mu_{b}=\beta \mu_{s}+\gamma \mu_{c}-\mu_{b},\;s\in \partial \Omega , \end{array} \end{array}$$
where
(69)$$\begin{array}{c} \begin{array}{c} M_{b}^{\left(1\right)}=-k_{b}\frac{Q_{2}\phi -Q_{1}\left(1-\phi \right)}{\ln\left(Q_{2}\phi \right)-\ln\left(Q_{1}\left(1-\phi \right)\right)},\;M_{b}^{\left(2\right)}\geq 0,\;M_{c}^{\left(1\right)}\geq 0,\\ M_{c}^{\left(2\right)}\geq 0,\;M_{s}^{\left(1\right)}=-k_{s}\frac{Q_{2}\psi -Q_{1}\left(1-\psi \right)}{\ln\left(Q_{2}\psi \right)-\ln\left(Q_{1}\left(1-\psi \right)\right)},\;M_{s}^{\left(2\right)}\geq 0, \end{array} \end{array}$$

$k_{b}$ and $k_{s}$ are two dimensionless parameters measuring the reaction rates in the bulk and the surface, respectively. $M_{b}^{\left(1\right)}$ and $M_{s}^{\left(1\right)}$ are two Logarithmic means, which depend on the order parameters and are non-negative. For more details about variable mobilities $M_{b}^{\left(1\right)}$ and $M_{s}^{\left(1\right)}$, readers are referred to our forthcoming paper [49]. The total energy of the system is dissipative. In the following, we briefly derive the model.

The dynamical description for the reactive transport in the binary system in bulk can be written as
(70)$$\begin{array}{c} \begin{array}{c} \partial_{t}\Phi =-\Lambda \left(\frac{\delta F}{\delta \Phi }-L1\right), \end{array} \end{array}$$
where $\Phi ={\left(\phi_{1},\phi_{2}\right)}^{T}$ is the volume fraction vector, $\phi_{i},i=1,2$ represent the volume fraction for the *i*th material component, $\Lambda =\Lambda^{\left(1\right)}-\Lambda^{\left(2\right)}\nabla^{2}$, $\Lambda^{\left(1\right)}$ is the mobility for reaction and $-\Lambda^{\left(2\right)}\nabla^{2}$ is the mobility for transport. The bulk free energy is given by
(71)$$\begin{array}{c} \begin{array}{c} F=\int_{\Omega }\left[\frac{b^{2}}{12}\nabla \phi_{1}\cdot \nabla \phi_{1}+\frac{b^{2}}{12}\nabla \phi_{2}\cdot \nabla \phi_{2}+\frac{1}{n_{1}}\phi_{1}\left(\ln\phi_{1}-1-\ln Q_{1}\right)+\frac{1}{n_{2}}\phi_{2}\left(\ln\phi_{2}-1-\ln Q_{2}\right)+\chi \phi_{1}\phi_{2}\right]dx. \end{array} \end{array}$$
In this model, we assume $n_{1}=n_{2}=n$. For reaction dynamics, the mobility operator is given by
(72)$$\begin{array}{c} \begin{array}{c} \Lambda^{\left(1\right)}=k_{b}\frac{Q_{2}\phi_{1}-Q_{1}\phi_{2}}{\ln\left(Q_{2}\phi_{1}\right)-\ln\left(Q_{1}\phi_{2}\right)}\left(\begin{array}{cc} 1 & -1\\ -1 & 1 \end{array}\right). \end{array} \end{array}$$
For transport dynamics, we set
(73)$$\begin{array}{c} \begin{array}{c} \Lambda^{\left(2\right)}=\left(\begin{array}{cc} \lambda_{11} & \lambda_{12}\\ \lambda_{12} & \lambda_{22} \end{array}\right). \end{array} \end{array}$$
The bulk transport equation is then given by
(74)$$\begin{array}{cc} \frac{\partial }{\partial t}\left(\begin{array}{c} \phi_{1}\\ \phi_{2} \end{array}\right) & =-k_{b}\frac{Q_{2}\phi_{1}-Q_{1}\phi_{2}}{\ln\left(Q_{2}\phi_{1}\right)-\ln\left(Q_{1}\phi_{2}\right)}\left(\begin{array}{cc} 1 & -1\\ -1 & 1 \end{array}\right)\left(\begin{array}{c} \mu_{1}-L\\ \mu_{2}-L \end{array}\right)+\left(\begin{array}{cc} \lambda_{11} & \lambda_{12}\\ \lambda_{12} & \lambda_{22} \end{array}\right)\nabla^{2}\left(\begin{array}{c} \mu_{1}-L\\ \mu_{2}-L \end{array}\right)\\ & =-k_{b}\frac{Q_{2}\phi_{1}-Q_{1}\phi_{2}}{\ln\left(Q_{2}\phi_{1}\right)-\ln\left(Q_{1}\phi_{2}\right)}\left(\begin{array}{cc} 1 & -1\\ -1 & 1 \end{array}\right)\left(\begin{array}{c} \mu_{1}\\ \mu_{2} \end{array}\right)+\left(\begin{array}{cc} \lambda_{11} & \lambda_{12}\\ \lambda_{12} & \lambda_{22} \end{array}\right)\nabla^{2}\left(\begin{array}{c} \mu_{1}-L\\ \mu_{2}-L \end{array}\right). \end{array}$$
Substituting $\mu_{1}=\frac{\delta F}{\delta \phi_{1}}=-\frac{b^{2}}{6}\nabla^{2}\phi_{1}+\frac{1}{n}\left(\ln\phi_{1}-\ln Q_{1}\right)+\chi \phi_{2},\mu_{2}=\frac{\delta F}{\delta \phi_{2}}=-\frac{b^{2}}{6}\nabla^{2}\phi_{2}+\frac{1}{n}\left(\ln\phi_{2}-\ln Q_{2}\right)+\chi \phi_{1}$ into the governing equation in the bulk, and using incompressible condition $\phi_{1}+\phi_{2}=1$, we obtain
(75)$$\begin{array}{c} \begin{array}{c} L=\frac{\left(\lambda_{11}+\lambda_{12}\right)\mu_{1}+\left(\lambda_{12}+\lambda_{22}\right)\mu_{2}}{\lambda_{11}+2\lambda_{12}+\lambda_{22}},\\ \frac{\partial \phi_{1}}{\partial t}=-k_{b}\frac{Q_{2}\phi_{1}-Q_{1}\phi_{2}}{\ln\left(Q_{2}\phi_{1}\right)-\ln\left(Q_{1}\phi_{2}\right)}\left(\mu_{1}-\mu_{2}\right)+\frac{\lambda_{11}\lambda_{22}-\lambda_{12}\lambda_{12}}{\lambda_{11}+2\lambda_{12}+\lambda_{22}}\nabla^{2}\left(\mu_{1}-\mu_{2}\right), \end{array} \end{array}$$
where the second equation is in the form: $\frac{\partial \phi }{\partial t}=-M_{b}^{\left(1\right)}\mu_{b}+M_{b}^{\left(2\right)}\nabla^{2}\mu_{b}$ when $\phi =\phi_{1}$ is identified and
(76)$$\begin{array}{c} \begin{array}{c} M_{b}^{\left(1\right)}=k_{b}\frac{Q_{2}\phi -Q_{1}\left(1-\phi \right)}{\ln\left(Q_{2}\phi \right)-\ln\left(Q_{1}\left(1-\phi \right)\right)},\\ M_{b}^{\left(2\right)}=\frac{\lambda_{11}\lambda_{22}-\lambda_{12}\lambda_{12}}{\lambda_{11}+2\lambda_{12}+\lambda_{22}},\\ \mu_{b}=\mu_{1}-\mu_{2}. \end{array} \end{array}$$

For the reactive binary system, we could introduce different order parameters besides $\phi =\phi_{1}$. For example, $\phi =\phi_{2}$ or $\phi =\phi_{1}-\phi_{2}$ in the bulk are all acceptable. Analogously, we obtain the mobilities/dynamics on the boundary. The choice of the order parameters on the boundary may be different from the one in the bulk. For example, $\phi_{s}=\left(\phi_{s,1}-\phi_{s,2}\right){|}_{\partial \Omega },\psi_{s}=\psi_{s,1}$, where $\psi_{s,1,2}$ are surface area fraction for the two polymeric components, respectively.

Recall the lattice model for polymers melts/solutions/blends [50], where one divides the domain into small lattices and uses the outermost lattices in the domain to represent the boundary. ϕ(x) at these outermost lattice can be identified as cϕ(s),s∈∂Ω, where *c* is a parameter to make sure ϕ(s) represents the area fraction while ϕ(x) is the volume fraction. We also add one layer of lattices at the outside of the domain and introduce ψ(s) to this layer. Although ϕ(s) and ψ(s) are in different layers, both of them can be viewed as representing quantities “on the surface”, when the thickness of the outermost lattices in the domain and outside is small enough. For any lattice we discussed above, we assume they are infinitesimal in macroscopic scale so that the “area fraction” and the volume fraction in the infinite thin layer limit can indeed be different.

There have been quite a number of papers investigating the reaction-diffusion phenomenon with dynamic boundary conditions in the last two decades in both experiments and modeling [51,52,53,54,55]. However, some of them are not thermodynamically consistent and the concept of the free energy used in the models is not clearly defined. In this paper, we use a simple binary reactive transport to illustrate how thermodynamically consistent models can be derived following the generalized Onsager principle. For more general reactive transport systems, readers are referred to our forthcoming paper [49].

## 4. Numerical Results for a Binary Reactive System

We take the binary reactive transport system as our numerical example to showcase how boundary dynamics impact bulk ones in a compact domain in 2-D. In the rectangular domain, the four sides are labeled as $\Gamma_{1},\Gamma_{2},\Gamma_{3}$, and $\Gamma_{4}$, respectively. We use the same free energies as those in the reactive transport system for the binary polymer system presented in section Section 3.4. The coupled dynamic equations in the bulk and at the boundary are given by the following
(77)$$\begin{array}{c} \begin{array}{c} \frac{\partial \phi }{\partial t}=-\left(M_{b}^{\left(1\right)}-M_{b}^{\left(2\right)}\nabla^{2}\right)\mu_{b},\;x\in \Omega ,\\ \mu_{b}=-\frac{b^{2}}{3}\nabla^{2}\phi +\frac{1}{n}\left(\ln\phi -\ln Q_{1}\right)-\frac{1}{n}\left(\ln\left(1-\phi \right)-\ln Q_{2}\right)+\chi \left(1-2\phi \right),\;x\in \Omega ,\\ \frac{\partial \phi }{\partial t}=-\left(M_{c}^{\left(1\right)}-M_{c}^{\left(2\right)}\nabla_{s}^{2}\right)\mu_{c}-\gamma M_{b}^{\left(2\right)}n\cdot \nabla \mu_{b},\;\mu_{c}=\frac{b^{2}}{6}n\cdot \nabla \phi -\eta \left(H\left(\psi \right)-\phi \right),\;s\in \Gamma_{1},\\ \frac{\partial \psi }{\partial t}=-\left(M_{s}^{\left(1\right)}-M_{s}^{\left(2\right)}\nabla_{s}^{2}\right)\mu_{s}-\beta M_{b}^{\left(2\right)}n\cdot \nabla \mu_{b},\;s\in \Gamma_{1},\\ \mu_{s}=-\frac{b^{2}}{3}\nabla_{s}^{2}\phi_{s}+\frac{1}{n}\left(\ln\psi -\ln Q_{1}\right)-\frac{1}{n}\left(\ln\left(1-\psi \right)-\ln Q_{2}\right)+\chi_{s}\left(1-2\psi \right)+\eta H^{\prime }\left(\psi \right)\left(H\left(\psi \right)-\phi \right),\;s\in \Gamma_{1},\\ \alpha M_{b}^{\left(2\right)}n\cdot \nabla \mu_{b}=\beta \mu_{s}+\gamma \mu_{c}-\mu_{b},\;s\in \Gamma_{1},\\ n\cdot \nabla \phi =n\cdot \nabla \mu_{b}=0,\;s\in \Gamma_{2},\Gamma_{3},\Gamma_{4}, \end{array} \end{array}$$
where $M_{b}^{\left(1\right)}=-k_{b}\frac{Q_{2}\phi -Q_{1}\left(1-\phi \right)}{\ln\left(Q_{2}\phi \right)-\ln\left(Q_{1}\left(1-\phi \right)\right)}$, $M_{s}^{\left(1\right)}=-k_{s}\frac{Q_{2}\psi -Q_{1}\left(1-\psi \right)}{\ln\left(Q_{2}\psi \right)-\ln\left(Q_{1}\left(1-\psi \right)\right)}$, $k_{b},k_{s},Q_{1},Q_{2}$ are assumed dimensionless parameters. In this model, we allow dynamic, reactive boundary conditions at $\Gamma_{1}$, and homogeneous Neumann boundary conditions at the other sides of the boundary. We adopt H(ψ)=ψ in the simulation and note that $k_{b}$ and $k_{s}$ parameterize the reactive rates in the bulk and boundary, respectively.

In the numerical treatment of the coupled PDE system, we use the energy quadratization technique coupled with the Crank–Nicolson method in time with time step $\delta t=1\times 10^{-5}$, and the second order finite difference method on staggered grids in space $\Omega ={\left[0,\;1\right]}^{2}$ with spatial mesh size 1/256 in both x and y direction to obtain a thermodynamically consistent numerical algorithm. The numerical algorithm guarantees that the total energy dissipates in time. For more details about the algorithm please refer to our previous papers [7,56,57]. We use the following initial condition in the simulations
(78)$$\begin{array}{c} \begin{array}{c} \phi \left(x\right)=0.3+0.01*\zeta ,\;\phi \left(s\right)=\lim_{x\to \Gamma_{1}}\phi \left(x\right),\;\psi \left(s\right)=\phi \left(s\right),\;\zeta \;\mathrm{is}\;\mathrm{random}\in \left[-1,1\right]. \end{array} \end{array}$$
Note that the Robin boundary condition on $\Gamma_{1}$ reduces to the homogeneous Neumann boundary condition when α→∞.

The numerical examples presented next are intended to showcase the effect of boundary reactive dynamics on the bulk dynamics at a set of parameter values. To explore the solution behavior of the model, a comprehensive study is needed which we will defer to another study. In the first simulation, we show the time evolution of spinodal decomposition of the binary polymer system with homogeneous Neumann boundary conditions at all four sides in Figure 1a–c. In the second simulation, we set the homogeneous Neumann boundary condition at $\Gamma_{i},i=2,3,4$ and the dynamic boundary condition at $\Gamma_{1}$ with $\alpha =\beta =\gamma =1,k_{s}=0,\eta =100$ and present the solution in Figure 1d–f. In the third one, we use $\alpha =\beta =\gamma =1,k_{s}=5\times 10^{-2},\eta =100$ with the same set of boundary conditions as in the second simulation and show the solution in Figure 1g–i. The first case decouples the bulk dynamics from the surface ones. The second case includes the coupling between the surface dynamics and the bulk dynamics without surface reaction. The third case includes reactive surface dynamics in dynamic coupling. Comparing the three simulations, we find that the dynamic boundary condition on the surface affects the bulk dynamics noticeably due to the non-negligible volume fraction flux (inward or outward) across the surface. The enhanced surface reaction on the boundary accelerate the merging of the droplets in the bulk in Figure 1g–i.

In this reactive-transport model, the volume fraction flux dictates boundary dynamics, as well as bulk ones near the boundary. We then depict the time evolution of the bulk volume fraction and bulk free energy in the three cases, respectively, in Figure 2. We notice that the bulk volume fraction with the homogeneous Neumann boundary condition is constant with time as expected, whereas the bulk volume fractions in the other two cases increase with time. The increasing bulk volume fractions in case 2 and case 3 lead to two different patterns in Figure 1d,i. In addition, the bulk free energies with dynamic boundary conditions dissipate faster than that with the homogeneous Neumann boundary condition. The enhanced reaction process accelerates the energy dissipation rate in the bulk.

Notice that the coupling energy at the boundary is a new physical quantity in this formulation and an intriguing one that determines the difference between ϕ(s) and ψ(s) and possibly influences transport dynamics in the bulk. Next, we investigate the effect of the coupling free energy together with the boundary reactive dynamics on the bulk dynamics at the presence of the inward/outward flux. The coupling free energy density adopted in the simulations is given by
(79)$$\begin{array}{c} e_{c}=\frac{\eta }{2}{\left(\psi -\phi \right)}^{2}, \end{array}$$
which measures the deviation between the two surface-bound variables. We present the case where the inward/outward flux is present (α=β=γ=1) in Figure 3. If $\eta =1\times 10^{2}$, representing a strong coupling, ψ(s) can significantly affect ϕ(s) as shown in Figure 3a. When η=0 in Figure 3b, the coupling energy vanishes so does the boundary dynamic effect of ψ(s). We also show the effect of surface reaction rate $k_{s}$ on bulk patterns in Figure 3b,d with $\eta =1\times 10^{2}$ and η=0, respectively. Figure 3a,b plot the numerical results without the surface reaction (i.e., $k_{s}=0$), while Figure 3c,d do the ones with the surface reaction ($k_{s}=5\times 10^{-2}$). Comparing Figure 2a,c (or Figure 2b,d), we find that the magnitude of the chemical reaction at the boundary, measured by the reactive rate parameter, affects the bulk dynamics near the reactive boundary significantly.

We then examine time evolution of the bulk volume fraction for these four test cases in Figure 4a. We observe that the bulk volume fraction increases with time in test 1, 3, and 4, which explain the mechanism of merging droplets or lamellar in Figure 3a,c,d. In Figure 4a, the bulk volume fraction in test 2 decreases with time, which indicates the bulk volume fraction flows from the bulk to the boundary so that there are less droplets near the boundary in Figure 3b. We also examine time evolution of the total free energies in the tests in Figure 4b, which show that our numerical algorithm used is energy-dissipative and warrant the second law of thermodynamics numerically.

To explore the effects of η and $k_{s}$ further, we employ another set of α,β, and γ values next. Firstly, we set α=0.1,β=γ=10 and plot the snapshots of the numerical solution of ϕ at T=8 with respect to several combinations of $\left(\eta ,k_{s}\right)$ in Figure 5. Secondly, we set α=10,β=γ=0.1 and plot the snapshots of the numerical solution of ϕ at T=8 with respect to the same combination of $\left(\eta ,k_{s}\right)$ in Figure 6. There are almost no differences among the snapshots in Figure 5 with respect to different η and $k_{s}$. Coincidentally, there are also no significant differences among the snapshots in Figure 6 with respect to different η and $k_{s}$ either. Finally, we show the time evolution of the bulk volume fractions in Figure 7. The results show the bulk volume fractions in tests 5–8 increase with time while the bulk volume fractions in tests 9–12 decrease with time. The inward/outward flux is influenced by the magnitudes of α,β,γ, and $k_{s},\eta$, which shows a slight effect on the pattern before T=8 in the eight tests.

## 5. Conclusions

Guided by the generalized Onsager principle, we have demonstrated a paradigm for deriving coupled, thermodynamically consistent dynamic models for the bulk and the boundary systematically using a phase field model for a binary fluid system as an example. All the existing, thermodynamically consistent phase field models for binary fluid systems accounting for dynamic boundary conditions are shown to be limits of this general phase field model. Then, we use a structure-preserving algorithm that we developed to numerically solve the phase field model for a binary fluid model with a reactive boundary to illustrate how boundary dynamics, including reactive dynamics and coupling energy, impact the bulk dynamics. This general framework anatomize the thermodynamical models with respect to both the bulk and boundary equations, laying a groundwork for analysing and designing efficient structure-preserving numerical approximations to these models in the future.

## Figures and Tables

**Figure 1 entropy-24-01683-f001:**
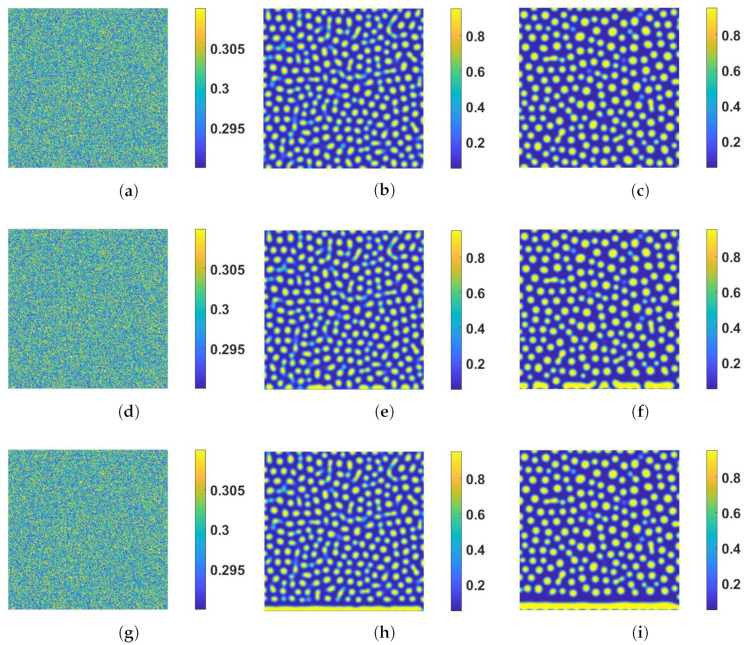
Time evolution of the spinodal decomposition of the binary polymer system with static (**a**–**c**) and dynamic boundary conditions (**d**–**i**), respectively. Snapshots of numerical solution ϕ are taken at T=0,4,8, respectively. (**a**–**c**). The solution is obtained with homogeneous Neumann boundary conditions on $\Gamma_{i},i=1,2,3,4$. (**d**–**f**). The solution is obtained with a set of dynamic boundary conditions at $k_{s}=0,\eta =100,\alpha =1,\beta =1,\gamma =1$. Dynamic boundary effects on the bulk solution near $\Gamma_{1}$ are observed. (**g**–**i**). The solution is obtained with a set of dynamic boundary conditions at $k_{s}=5\times 10^{-2},\eta =100,\alpha =1,\beta =1,\gamma =1$. The boundary reactive dynamics shown to impact the bulk solution more significantly near $\Gamma_{1}$. The other model parameter values used in the simulations are given as follows: $\beta =\gamma =n=Q_{1}=Q_{2}=1,M_{b}^{\left(1\right)}=M_{c}^{\left(1\right)}=0,M_{b}^{\left(2\right)}=M_{c}^{\left(2\right)}=M_{s}^{\left(2\right)}=1\times 10^{-4},\chi =\chi_{s}=4,k_{b}=0,b=0.02$.

**Figure 2 entropy-24-01683-f002:**
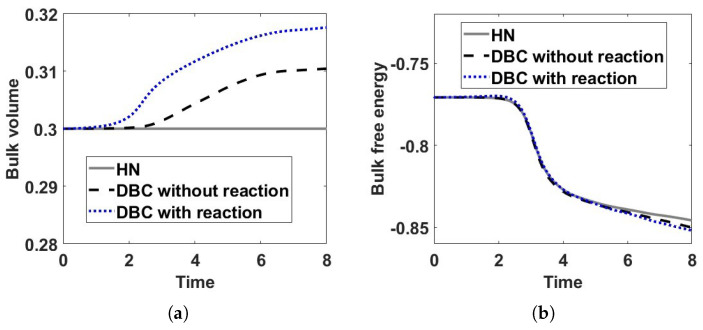
Time evolution of the bulk volume fraction in (**a**) and bulk free energy in (**b**) with respect to the three simulations depicted in Figure 1. It is obvious that the bulk volume fraction with homogeneous Neumann boundary conditions keeps as a constant. However, no matter whether or not there is the reaction on the boundary, the bulk volume fraction increases with time when the boundary conditions are dynamic. Compared with the DBCs without reaction, the bulk volume fraction in the DBCs with reaction increases faster than that without reaction. The bulk free energy in each case decays in time as expected. Due to the existence of chemical reaction, the bulk free energy in the third simulation (with surface reaction) decreases faster than that in the second one (without surface reaction).

**Figure 3 entropy-24-01683-f003:**
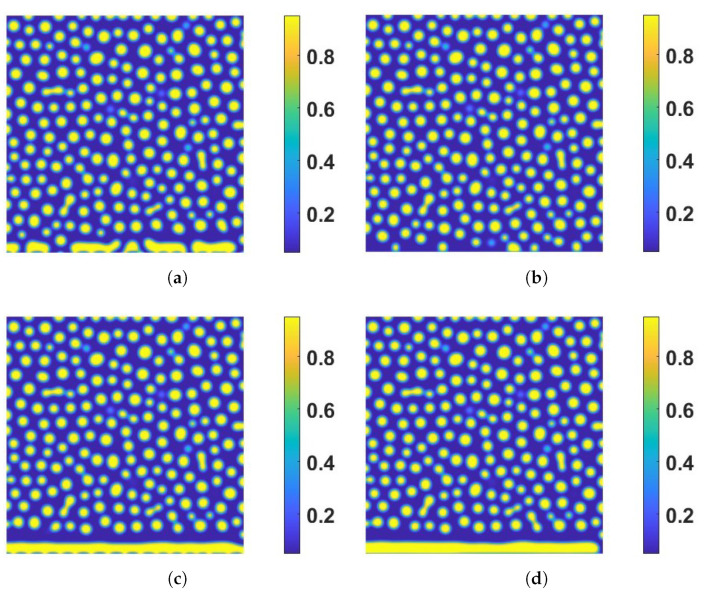
The effect of the boundary reaction rate and magnitude of the coupling free energy on bulk dynamics in the presence of inward/outward flux (α=β=γ=1). Snapshots of the solution of ϕ are taken T=8. (**a**) Test 1: strong coupling without boundary reaction ($k_{s}=0,\eta =1\times 10^{2}$). (**b**) Test 2: decoupling without boundary reaction ($k_{s}=\eta =0$). (**c**) Test 3: strong coupling with boundary reaction ($k_{s}=5\times 10^{-2},\eta =1\times 10^{2}$). (**d**) Test 4: decoupling with boundary reaction ($k_{s}=5\times 10^{-2},\eta =0$). The other model parameter values are the same as those listed in the caption in Figure 1. These four plots show that the non-trivial effect from both the coupling energy and the boundary reaction impacts the bulk dynamics nearby the boundary. In (**a**,**c**, and **d**), there are large merging droplets or lamellar shown near the boundary.

**Figure 4 entropy-24-01683-f004:**
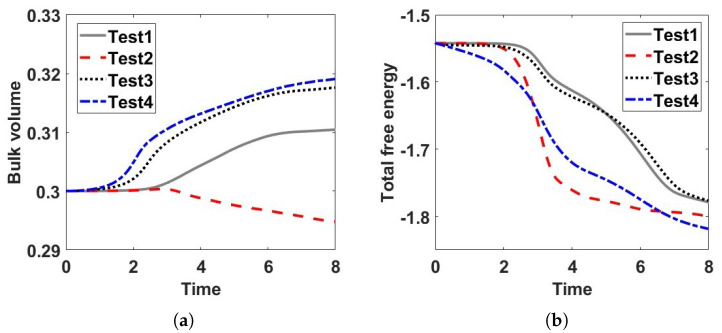
Time evolution of the bulk volume fractions in (**a**) and the total free energy in (**b**) with respect to the four tests in Figure 2. We find the bulk volume fractions increase with time in tests 1, 3, and 4, which are related to the merging droplets or lamellar in Figure 3a,b,d. Since there is a leakage of bulk volume fraction in test 2, we do not find any large droplets or lamellar in Figure 3b. The total free energy in each test decays in time as expected.

**Figure 5 entropy-24-01683-f005:**
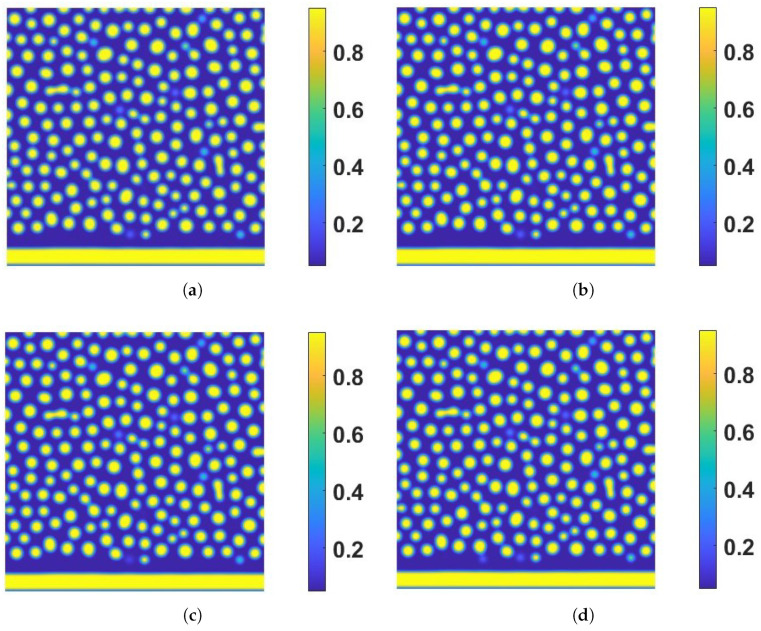
The effect of the boundary reaction rate and magnitude of the coupling free energy on bulk dynamics in the presence of inward/outward flux (α=0.1,β=γ=10). Snapshots of the solution of ϕ are taken T=8. (**a**) Test 5: strong coupling without boundary reaction ($k_{s}=0,\eta =1\times 10^{2}$). (**b**) Test 6: decoupling without boundary reaction ($k_{s}=\eta =0$). (**c**) Test 7: strong coupling with boundary reaction ($k_{s}=5\times 10^{-2},\eta =1\times 10^{2}$). (**d**) Test 8: decoupling with boundary reaction ($k_{s}=5\times 10^{-2},\eta =0$). The other model parameter values are the same as those listed in the caption in Figure 1. These four plots show both the coupling energy and the boundary reaction impact the bulk dynamics near the boundary.

**Figure 6 entropy-24-01683-f006:**
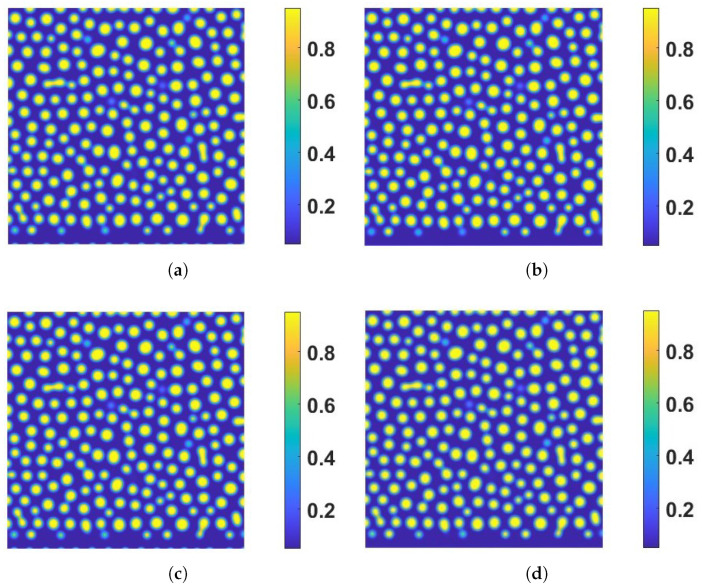
The effect of the boundary reaction rate and magnitude of the coupling free energy on bulk dynamics in the presence of inward/outward flux (α=10,β=γ=0.1). Snapshots of the solution of ϕ are taken T=8. (**a**) Test 9: strong coupling without boundary reaction ($k_{s}=0,\eta =1\times 10^{2}$). (**b**) Test 10: decoupling without boundary reaction ($k_{s}=\eta =0$). (**c**) Test 11: strong coupling with boundary reaction ($k_{s}=5\times 10^{-2},\eta =1\times 10^{2}$). (**d**) Test 12: decoupling with boundary reaction ($k_{s}=5\times 10^{-2},\eta =0$). The other model parameter values are the same as those listed in the caption in Figure 1. These four plots show that both the coupling energy and the boundary reaction impact the bulk dynamics near the boundary.

**Figure 7 entropy-24-01683-f007:**
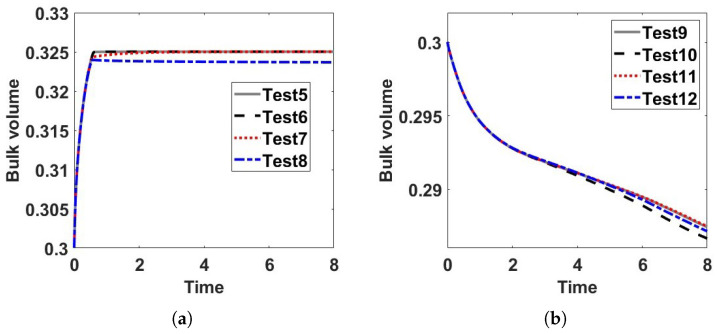
Time evolution of the bulk volume fractions in test 5–8 in (**a**) and test 9–12 in (**b**). We find the bulk volume fractions increase with time in tests 5–8, which are related to the merging droplets or lamellar in Figure 5. Since there are leakages of bulk volume fractions in tests 9–12, we find there are lamellar formed by the other phase in Figure 6.

## Data Availability

Not applicable.

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
