# Peer review of "Thermodynamically Consistent Models for Coupled Bulk and Surface Dynamics"

_entropy, 2022, doi:10.3390/e24111683_

Round 1
Reviewer 1 Report
In the manuscript ``Thermodynamically Consistent Models for Coupled Bulk and Surface Dynamics'', Jing and Wang presented a constructive paradigm to derive thermodynamically consistent models, with a focus on the coupling between the bulk and surface dynamics. The derivation was based on the generalized Onsager principle. The model was presented in detail and a phase field model was used for illustration. The manuscript was written well and the results were presented clearly. I mostly would like to see some clarifications in the manuscript.
2.1:
I would suggest not calling $\mu$ as chemical potential because it might cause confusion. As in standard physical chemistry it is the gradient of chemical potential that drives the transport.
2.2:
* In the part when the author discussed the reason for using two phase variables, I don't understand the surface part (the definitions of $s_i$ and $N_i^s$). Does the surface have a finite thickness? How to determine if one particle with finite volume $v_i$ is on the surface?
* The author stated that "there are no reasons to believe $N_i=N_i^s$ nor $v_2/v_1 = s_2/s_1$", but they must be related in some ways, right?
* Would the authors provide some physical/real examples for the coupling free energy (2.7)?
* In the line of $\mu_c$ in Eq. (2.10), is it necessary to include $e_s$ in all terms? Since $e_s$ is only a function of $\psi$, its partial derivative w.r.t. $\phi$ would be zero.
3:
Could the author provide some introduction to the JW, KL, LW, and KS models, for the benefit of non-expert?
3.4:
Could the authors explain the specific form of mobility (3.31)?
Author Response
Please see the attached response to referee's report.

Reviewer 2 Report
In this paper, the authors propose a scheme coupling the dynamics of the bulk of a thermodynamic system to that of its boundaries. This is done by enforcing a generalized Onsager principle. The scheme is applied to models of binary materials and some special situations are recovered within the general framework. Finally, numerical results are presented for a specific model of binary reacting fluids.
The paper is appropriate for the journal although it is rather dense and difficult to read. I recommend acceptance but suggest the following points:
- The list of references should be ordered by citation order rather than by alphabetically.
- A few misprints: "there are an abundance"->"there is an abundance", "There have been a surge"->"There has been a surge", "different order parameter beside"->"different order parameter besides".
- The acronyms in Sec. 3 (JW, KL, ...) should be spelled out (Jing-Wan, Knopf-Lam, ...).
- In the list of references,
o Names as Cahn, Hilliard, Wentzell, Allen, Thomson, Onsager, ... should be capitalized.
o First names of the authors (Howard, Chun-Wei, Pierluigi, ...) should be replaced by initials (H., C.-W., P., ...).
o Names of journals (Physical Review E, ...) should be abbreviated (Phys. Rev. E, ...) and capitalized.
o Replace Ref. 9 by DOI=10.3233/ASY-201646.
o Replace Ref. 25 by DOI=10.1088/1361-6544/ab8351.
o Replace Ref. 26 by DOI=10.1051/m2an/2020090.
o Replace Ref. 55 by DOI=10.4310/CMS.2020.v18.n5.a11.
Author Response
Please see the response to the referee's report letter.
